# An mRNA-display derived cyclic peptide scaffold reveals the substrate binding interactions of an N-terminal cysteine oxidase

Yannasittha Jiramongkol [1,2], Karishma Patel [1,3], Jason Johansen-Leete[1], Joshua W. C. Maxwell[1,4], Yiqun Chang[5], Jonathan J. Du [5], Toby Passioura[6], Kristina M. Cook [7], Richard J. Payne [1,4] & Mark D. White [1] ✉

N-terminal cysteine oxidases (NCOs) act as enzymatic oxygen ($O_2$) sensors, coordinating cellular changes to hypoxia in animals and plants. They regulate the $O_2$-dependent stability of proteins bearing an N-terminal cysteine residue through the N-degron pathway. Despite their important role in hypoxic adaptation, which renders them potential therapeutic and agrichemical targets, structural information on NCO substrate binding remains elusive. To overcome this challenge, we employed a unique strategy by which a cyclic peptide inhibitor of the mammalian NCO, 2-aminoethanethiol dioxygenase (ADO), was identified by mRNA display and used as a scaffold to graft substrate moieties. This allowed the determination of two substrate analogue-bound crystal structures of ADO. Key binding interactions were revealed, including bidentate coordination of the N-terminal residue at the metal cofactor. Subsequent structure guided mutagenesis identified aspartate-206 as an essential catalytic residue, playing a role in reactive oxygen intermediate orientation or stabilisation. These findings provide fundamental information on ADO substrate interactions, which can elucidate enzyme mechanism and act as a platform for chemical discovery.

Molecular oxygen ($O_2$) is a vital biological resource, delivery of which is impaired in many diseases due to inadequate blood perfusion[1]. As a result, $O_2$ deprivation (hypoxia) contributes to some of the leading causes of death and disability in developed countries, including stroke, ischaemic heart disorders, and cancer[2]. To help address these therapeutic challenges, the molecular mechanisms underpinning $O_2$ homoeostasis and hypoxic adaptation have been targeted to beneficially alter low $O_2$ stress responses. This strategy has been successfully implemented in the treatment of anaemia[3,4] and renal cell cancer[5]. Here, FDA approved drugs of the hypoxia inducible factor (HIF) system, which coordinates widespread transcriptional changes to low $O_2$ in metazoans[6], have been developed. Nevertheless, additional processes that sense and respond to hypoxia have recently been identified, providing alternative approaches to combat issues associated with $O_2$ deprivation and hypoxic stress. This includes the cysteine (Cys) branch of the N-degron pathway, which controls the

[1]School of Chemistry, The University of Sydney, Sydney, NSW, Australia. [2]Faculty of Science, Charles Perkins Centre, The University of Sydney, Sydney, NSW, Australia. [3]School of Life and Environmental Sciences, The University of Sydney, Sydney, NSW, Australia. [4]Australian Research Council Centre of Excellence for Innovations in Peptide and Protein Science, The University of Sydney, Sydney, NSW, Australia. [5]School of Pharmacy, The University of Sydney, Sydney, NSW, Australia. [6]Sydney Analytical Core Research Facility, The University of Sydney, Sydney, NSW, Australia. [7]Faculty of Medicine and Health, Charles Perkins Centre, The University of Sydney, Sydney, NSW, Australia. ✉e-mail: mark.white@sydney.edu.au

$O_2$-dependent stability of proteins bearing a co- or post-translationally exposed N-terminal (Nt-) Cys in both plants and animals[7].

The N-degron pathway is a universal and multifaceted regulatory system, which dictates the half-life of a protein based on the identity and modification state of its Nt-amino acid[8]. In higher eukaryotes, Nt-Cys acts as a tertiary destabilising residue, which promotes proteasomal degradation following three sequential modifications: Nt-Cys oxidation, Nt-arginylation and ubiquitinion (Supplementary Fig. 1). The first modification, Nt-Cys oxidation, is enzymatically regulated by Nt-Cys oxidases (NCOs), which use both atoms of $O_2$ to catalyse sulfinylation of the Nt-Cys residue[7,9,10]. NCOs are kinetically tailored to couple $O_2$ availability to enzyme activity and, consequently, protein levels through a high $K_m$ and low affinity for $O_2$[7,11]. Under normoxic conditions, NCOs constitutively mark their protein substrates for removal through the Cys branch of the N-degron pathway. However, during hypoxia, NCOs become inactivated, resulting in target stabilisation and cellular change.

In animals, the Cys branch of the N-degron pathway is regulated by the NCO 2-aminoethanethiol dioxygenase (ADO), which also plays a role in cysteamine catabolism and taurine biosynthesis[7,12]. Known ADO targets are predominantly associated with cell signalling events, such as regulators of G-protein coupled signalling (RGS) 4 and 5, and the atypical cytokine interleukin 32 (IL32), suggesting that ADO complements the transcriptional output of HIF by coordinating an earlier response to hypoxia[13]. RGS4 and 5 are implicated in maintaining healthy cardiovascular function while IL32 regulates pro-inflammatory cytokine networks and promotes angiogenesis[14,15]. RGS4, RGS5 and IL32 are also associated with various cancers, where they contribute to disease proliferation, invasiveness and migration, potentially making ADO a universal drug target for these conditions[16–21]. ADO, itself, has been linked to immune invasion and cancer cell redox homeostasis[22,23].

Biochemically, ADO is a non-haem iron-dependent thiol dioxygenase (TDO) from the cupin fold family of enzymes, consisting of a conserved double stranded β-helix (DSBH) core in which the active site is situated[24,25]. The catalytic centre comprises a ferrous iron ($Fe^{2+}$), which is octahedrally coordinated by a facial triad of three histidine residues, with three water molecules occupying the remaining ligation sites at rest. ADO retains low sequence and structural homology with cysteine dioxygenase (CDO), which processes free L-Cys as part of sulphur metabolism[26]. CDO coordinates L-Cys in a bidentate arrangement through ligation of the amine and thiol group, leaving one coordination site available for $O_2$ to bind in an end-on orientation[27–29]. This promotes formation a putative iron(III) superoxo intermediate, which reacts with the thiol through radical recombination (Supplementary Fig. 2)[30,31]. Whether the reaction proceeds through initial addition of the proximal or distal oxygen atom remains to be resolved, as no intermediates have been conclusively identified, although persulfenate species have been observed *in crystallo*[27,29].

Interestingly, ADO has a different distribution of amino acids in its active site relative to CDO, suggesting that it uses alternative strategies to bind and modify its substrates[24,25]. This is supported by spectroscopic studies on the metal centre, which suggest that cysteamine, the small molecule substrate of ADO, coordinates iron in a monodentate arrangement through ligation of the thiol group, leaving two sites available for $O_2$ to bind in a side-on orientation[32,33]. However, recent experiments contradict this view, indicating that a superoxo intermediate can form in a catalytically impaired substituent of ADO and that a free Nt-thiol, Nt-amine and active site metal are vital for protein substrate binding and turnover, both of which imply bidentate coordination[34–36].

No substrate bound crystal structure of ADO, or any NCO, has been obtained, owing to the rapid turnover and low affinity of native and analogue substrate sequences[36]. This has limited enzyme characterisation and rational manipulation through chemical discovery. To overcome these challenges, we employed Random nonstandard Peptide Integrated Display (RaPID) technology, a variation of mRNA display that allows the incorporation of non-standard amino acids through genetic reprogramming[37–40], to identify cyclic peptide inhibitors of ADO. These efforts resulted in the discovery of targeted chemical modulators of this enzyme, which exhibit competitive and uncompetitive inhibition modes based on their kinetic properties. One of the cyclic peptide inhibitors was then used as a scaffold to graft a series of substrate analogue motifs bearing a pseudo-Nt-Cys or -Ser residue, allowing two complex crystal structures of ADO to be determined at high resolution. In both structures the Nt-residue coordinates the metal centre in a bidentate arrangement, leaving one ligation site for $O_2$ to bind in an end-on orientation, analogous to CDO. Conserved active site residues were subsequently probed through mutagenesis, highlighting amino acids involved in binding and activity, with ADO-D206 identified as an essential catalytic residue. Given its position relative to the $O_2$ binding site, ADO-D206 may play a role in orientating, directing or stabilising a reactive oxygen intermediate, ensuring correct reaction with the substrate thiol. Together this work can help elucidate NCO mechanism and facilitate the rational development of new and improved ADO inhibitors to study and manipulate the Cys branch of the N-degron pathway in the context of hypoxic disease.

## Results

### The identification and characterisation of cyclic peptide inhibitors of ADO

To provide chemical tools to selectively probe and manipulate ADO activity, RaPID was used to identify cyclic peptide (CP) inhibitors of human ADO. A semi-randomised library of DNA was transcribed into RNA, ligated to puromycin, and translated in vitro to generate a pool of over $10^{12}$ CPs linked to their corresponding mRNA/cDNA 'barcodes' (Fig. 1a). Peptides were initiated with *N*-chloroacetyl-L-tyrosine to promote spontaneous cyclisation with a downstream Cys through thioether formation. This library was incubated with biotinylated ADO, immobilised on streptavidin beads, to pull out high affinity binding partners. Bound DNA sequences were recovered, amplified by PCR, and used as the starting material for additional selection rounds.

After six cycles, clear enrichment of several CPs was observed by next-generation sequencing. The most enriched CPs (CP1-8, numbered according to their percentage enrichment in the final selection round) were selected for subsequent analysis (Table 1). CP1, 5, 6 and 7 are related in sequence, while CP2, 3, 4, and 8 are unique compositions (Fig. 1b). These CPs were subsequently produced by solid-phase peptide synthesis (SPPS) and analysed by surface plasmon resonance (SPR) using single cycle kinetic (SCK) analysis to confirm and quantify binding to ADO. ADO could interact with each CP to varying degrees (Supplementary Fig. 4, Table 1, Fig. 1d *left*). CP1, 5, 6, and 8 exhibited the tightest interactions, generating equilibrium dissociation constants ($K_D$) of 35, 40, 66, and 5 nM, respectively, while CP2, 3, and 4 displayed significantly lower binding affinities, with $K_D$ values of >5000, >5000, and 1300 nM. Unfortunately, CP7 was not amenable to affinity quantification by SPR as it displayed non-specific binding at the concentrations tested, which resulted in a significantly slow off rate (Supplementary Fig. 4).

We next tested the ability of the CPs to inhibit ADO activity. ADO activity was measured with and without 10 µM of each CP using an established stopped assay in which the oxidation of a representative substrate ($RGS5_{2-15}$; a synthetic peptide corresponding to the methionine excised N-terminus of RGS5, residues 2 to 15) was monitored by liquid chromatography coupled mass spectrometry (LCMS)[7,35,36]. Every CP except CP3, which demonstrated the weakest $K_D$ by SPR, reduced ADO activity, with CP1, 5, 6, and 8 displaying the greatest impact on substrate turnover (Fig. 1C). Inhibition was subsequently quantified through half-maximal inhibitory concentration ($IC_{50}$) measurements, confirming that CP1, 5, 6, and 8 are the most

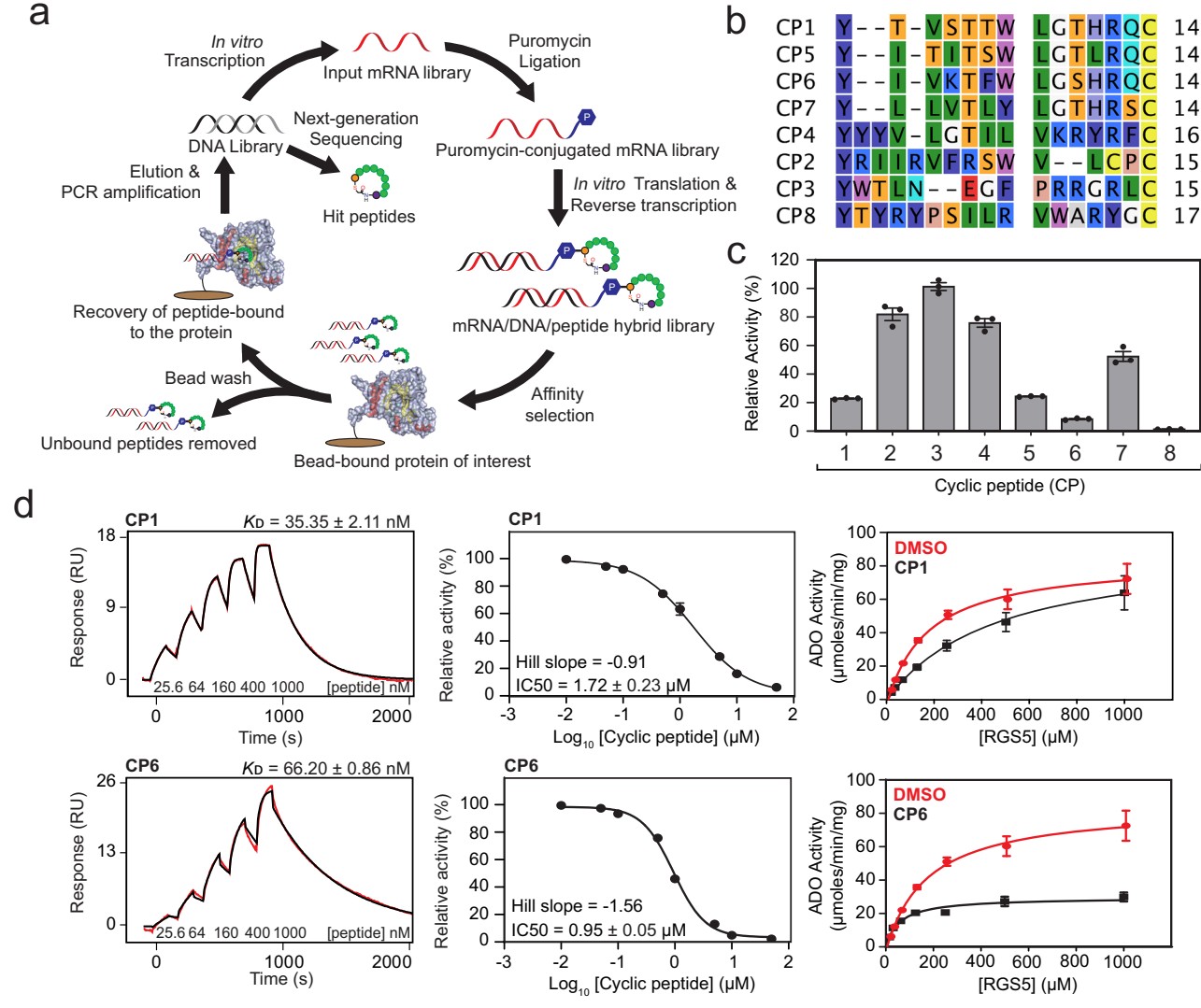

**Fig. 1 | The identification and characterisation of cyclic peptide (CP) inhibitors of ADO. a** A schematic highlighting the key steps of RaPID. Generated using Adobe Illustrator, Revvity Chemdraw and CCP4MG. Protein coordinates were obtained from the PDB using accession code 8UAN. **b** A sequence alignment of the top eight unique CPs discovered for ADO. **c** Single-dose inhibition assays: Relative enzymatic activity of ADO with RGS5$_{2-15}$ (100 μM) in the presence of CP (10 μM) at 37 °C. The average of three independent experiments ($n = 3$) is shown (error bars show the standard error). **d** Biophysical analysis and inhibitory responses of CP1 and CP6. (*Left*) Single-cycle kinetic (SCK) SPR sensorgrams of CP1 and CP6 with ADO. The sensorgrams are shown in red and the fits to the data are shown in black. The concentrations of CPs used in the titration and the equilibrium dissociation constants ($K_D$) are shown ($K_D$ given as the geometric mean of a minimum of three independent SPR measurements ($n = 3$) with standard error. (*Middle*) Dose-response curves with the half-maximal inhibitory concentration (IC$_{50}$) values for CP1 and CP6. The average of three independent experiments ($n = 3$) is shown (error bars show the standard error). (*Right*) Michaelis-Menten kinetic plots of ADO activity in the absence and presence of CP1 and CP6 performed under aerobic conditions at 37 °C. The average of three independent experiments ($n = 3$) is shown (error bars show the standard error). Source data are provided as Source Data file.

potent inhibitors, with IC$_{50}$ values in the single digit μM range (Supplementary Fig. 5, Table 1, Fig. 1d *middle*).

The inhibition modes of CP1, 5, 6, and 8 were initially assessed using a SPR-based competition assay, which monitored the binding of full-length RSG5 (RGS5$^{FL}$; which is ~10 times larger than the CPs) to ADO in the presence of different concentrations of CP. Each inhibitor could impede RGS5 association, as demonstrated by a decrease in response signal, suggesting that CP1, 5, 6, and 8 preferentially bind at the ADO active site compared to native substrate (Supplementary Fig. 6). The modes of inhibition were further characterised using the Michaelis Menten model of enzyme kinetics, by comparing the kinetic parameters of ADO with and without CP1, 5, 6, and 8. Each CP was added at its IC$_{50}$ concentration to capture changes in substrate association and turnover during partial inactivation of the enzyme (Supplementary Fig. 7, Supplementary Table 1, Fig. 1d *right*). This revealed that CP1 and

5 are competitive inhibitors, which increase the apparent $K_m$ of ADO, without altering $k_{cat}$, while CP6 and 8 exhibit the kinetic properties of uncompetitive inhibitors, decreasing both the apparent $K_m$ and $k_{cat}$. This was unexpected as CP1, 5, and 6 are related in sequence (Fig. 1b). However, CP6 contains two unique residues, K4 and F6, which may influence inhibitor behaviour.

Together this established targeted inhibitors of ADO, which exhibited distinct modes of action as demonstrated by changes in apparent substrate association and enzyme turnover.

## CP6 inhibits ADO by blocking the active site and detracting a putative catalytic residue

A crystal structure of ADO in complex with CP6 was obtained through co-crystallisation to characterise its binding mode and evaluate how differences in CP sequence can influence the means of inhibition.

**Table 1 | A summary of the cyclic peptides identified by RaPID and their biophysical and inhibitory properties**

| CP | Sequence | Enrichment (%) | $K_D$ (nM) | IC50 (µM) | Inhibition mode (apparent) |
|---|---|---|---|---|---|
| 1 | YTVSTTWLGTHRQC | 9.0 | 35 ± 2.1 | 1.72 ± 0.23 | Competitive |
| 2 | YRIIRVFRSWVLCPC | 4.3 | >5000 | >50 | - |
| 3 | YWTLNEGFPRRGRLC | 2.6 | >5000 | - | - |
| 4 | YYYVLGTILVKRYRFC | 2.0 | 1300 ± 140 | >50 | - |
| 5 | YITITSWLGTLRQC | 1.8 | 40 ± 0.80 | 5.68 ± 0.53 | Competitive |
| 6 | YIVKTFWLGSHRQC | 1.7 | 66 ± 0.86 | 0.95 ± 0.05 | Uncompetitive |
| 7 | YLLVTLYLGTHRSC | 1.5 | - | 9.92 ± 0.29 | - |
| 8 | YTYRYPSILRVWARYGC | 1.1 | 4.7 ± 0.35 | 1.73 ± 0.02 | Uncompetitive |

Values were calculated using data presented in Supplementary Figs. 4, 5 and 7. Cyclisation occurs between *N*-chloroacetyl-L-tyrosine, in position one, and the first cysteine in the sequence.

Initial attempts to crystallise CP6 in complex with wild-type ADO proved unsuccessful. However, two crystal structures of unbound ADO, which were published during this investigation, highlighted alterations that result in an amenable crystallisation construct, including metal cofactor substitution and surface Cys mutations[24,25]. Following these strategies, a crystal structure of cobalt-incorporated ADO in complex with CP6 was obtained at 1.74 Å resolution (Protein Data Bank [PDB] ID: 9DXU; Fig. 2, Supplementary Table 9, Supplementary Figs. 8–10).

CP6 adopts an antiparallel β-sheet conformation, which is maintained through a series of intramolecular hydrogen bonds between backbone and sidechain atoms (Fig. 2a, Supplementary Table 2). The peptide sits across one end of the DSBH core of ADO, blocking the active site entrance. CP6 binding is mediated by a number of intermolecular interactions, including pi-stacking between the sidechain of CP6-W7 and the sidechain of ADO-Y87, pi-stacking between the C-terminal amide of CP6-C14 and the sidechain of ADO-W257, and hydrogen bonds between the sidechain of CP6-R12 and the backbone carbonyls of ADO-Q252, -A253 and -F256 (Supplementary Fig. 8, Supplementary Table 3).

Globally, CP6 binding causes minor changes in ADO structure, with superimposition based on secondary structure matching resulting in a root mean squared deviation (RMSD) of 0.74 when compared to PDB:8UAN, a crystal structure of cobalt-incorporated ADO in the absence of a CP (Fig. 2b, Supplementary Fig. 9; 220 residues aligned)[34]. However, specific permutations are observed, including a shift in a hairpin loop (residues 212 to 220), which moves ~2 Å away from CP6 (Supplementary Fig. 9). Additional changes in ADO structure are largely restricted to amino acid sidechains including, most notably, ADO-W257, which rotates to stack with the C-terminal amide of CP6-C14, and ADO-E92, which adopts a distinct conformation to avoid clashing with CP6-F6 (Fig. 2b). Additional electron density is observed around the active site metal, which was modelled as Tris coordinating cobalt in multiple conformations (Supplementary Fig. 10). Tris is known to interact with metal ions and has been observed coordinating with the metal centre of plant cysteine oxidase 2 (PCO2), a structural and functional homologue of ADO[41,42]. Nevertheless, given the complexity of the density, other interpretations are possible.

As stated above, CP6 contains two unique residues, K4 and F6, which are the most likely candidates for reducing ADO turnover and conferring the kinetic characteristics of an uncompetitive inhibitor based on sequence alignments with CP1 and 5, which act as competitive inhibitors. CP6-K4 and -F6 interact with ADO-D215 and ADO-E92, respectively. While there is no immediate rationale for the functional contribution of the former, CP6-F6 promotes formation of a hydrogen bonding network down to the active site via ADO-E92, -H89, -Y87 and a water molecule, which is not present in the peptide-free structure. This pulls the side chain of ADO-D206, a putative catalytic residue, away from the metal centre (Fig. 2b)[43].

To confirm the importance of these interactions in reducing ADO turnover, two single amino acid substituents of CP6, CP6-K4A and

-F6S, were synthesised and assessed (Supplementary Fig. 11, Supplementary Table 4, Fig. 2c). While CP6-K4A had a large impact on CP and substrate association, increasing the $K_D$, IC$_{50}$ and apparent $K_m$ (relative to CP6), CP6-F6S had a significant effect on turnover, increasing the apparent $k_{cat}$ (relative to CP6) from 16 s⁻¹ to 38 s⁻¹ (Fig. 2c, Supplementary Table 4). A similar trend was observed when CP was added at two times the IC$_{50}$ concentration (Supplementary Fig. 12, Supplementart Table 5). The role of ADO-E92 was also assessed through mutagenesis. In the presence of ADO-E92A, CP6 exhibited characteristics that more closely align with a competitive inhibitor, increasing the apparent $K_m$ and $k_{cat}$ relative to the DMSO control and wild-type ADO in the presence of CP6, respectively (Supplementary Fig. 13, Supplementary Table 6).

Together, this suggests that CP6-F6 reduces turnover and instils the kinetic properties of an uncompetitive inhibitor by promoting the formation of an ADO-E92-mediated hydrogen bonding network down to the active site that pulls ADO-D206 into an unfavourable conformation. Accordingly, CP6 appears to inhibit ADO by both blocking the active site and detracting a putative catalytic residue.

### Employing CP6 as a scaffold to graft substrate moieties and determine ADO binding interactions

Structural information on ADO substrate binding has been elusive. Several attempts to crystallise ADO in complex with different substrate constructs and analogues proved unsuccessful due to product formation and low affinity, respectively, even following active site metal substitution. To overcome these issues, CP6 was used as a molecular scaffold to graft substrate moieties, anticipating that its binding and inhibition properties would enhance substrate complex formation by increasing association and reducing turnover.

CP6-L8 protrudes into the active site of ADO without contributing to binding, rendering it a viable site for substrate moiety installation (Fig. 3a, b, Supplementary Fig. 14). In the first instance, CP6-L8 was replaced with ornithine (Orn; O) or lysine (Lys), allowing the sidechain amine to be coupled to Cys or serine (Ser), the latter of which is an amenable substrate analogue[36]. This generated a pseudo-N-terminus with conformational and rotational flexibility, which could reach the ADO active site metal (a distance of 7.5 Å; Fig. 3, Supplementary Fig. 14). The association of these CP6-substituents was assessed by SPR using SCK, revealing that Cys and Ser conjugated to a Lys sidechain enhanced binding relative to unmodified CP6 (Supplementary Fig. 15, Fig. 3c). Encouraged by this data, co-crystallisation trials with CP6-L8K coupled to Cys and Ser were conducted, resulting in a crystal structure of cobalt-incorporated ADO in complex with CP6-L8K-Ser at 1.60 Å resolution (PDB:9DXV; Fig. 4, Supplementary Table 9). The Ser residue of CP6-L8K-Ser coordinates the metal centre in a bidentate arrangement through ligation of both the hydroxyl and amine group, leaving one water molecule trans to ADO-H112, which is the putative site of oxygen binding and activation. The water molecule trans to ADO-H112 also forms a hydrogen bond with ADO-D206, which points towards the metal centre through an additional interaction with the amine of CP6-

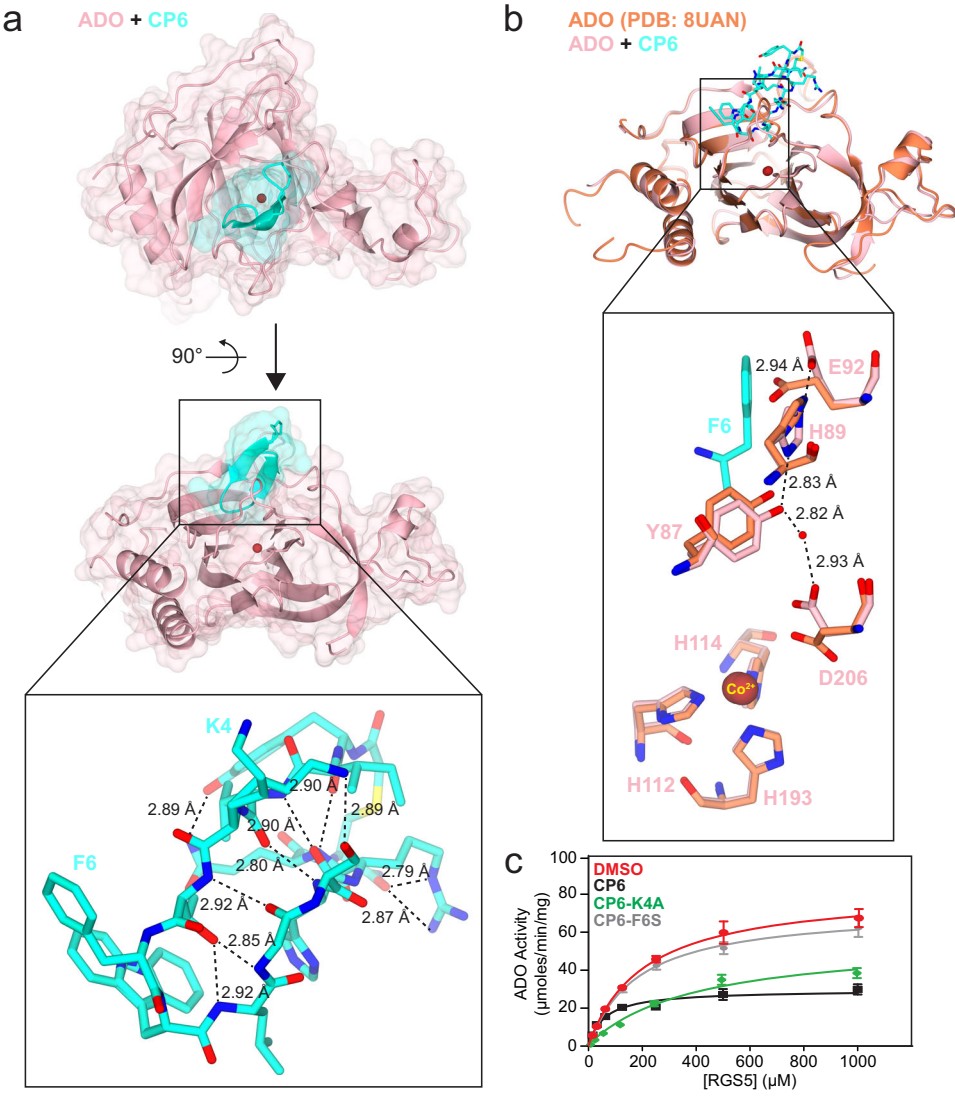

**Fig. 2 | A crystal structure of ADO in complex with CP6 elucidates its mode of inhibition and highlights a putative catalytic residue. a** The crystal structure of cobalt-incorporated ADO in complex with CP6 (1.74 Å resolution), displayed through ribbon, surface (transparent) and cylinder representations. CP6 (cyan) forms an antiparallel β-sheet, which lies across the DSBH of ADO (pink), blocking the active site. The conformation of CP6 is maintained through a series of intra-molecular hydrogen bonding interactions (boxed insert). **b** The superimposed crystal structures of CP6-bound (light pink) and CP-free (orange; PDB: 8UAN) cobalt-incorporated ADO. CP6-F6 promotes a hydrogen bond network through

ADO-E92, which pulls the sidechain of ADO-D206 away from the metal centre (boxed insert). **c** Michaelis-Menten kinetic plots of ADO in the absence and presence of CP6, CP6-K4A and CP6-F6S under aerobic conditions at 37 °C, which highlights the role of CP6-F6 in reducing ADO turnover. Each CP was added at its half-maximal inhibitory concentration (IC$_{50}$) to capture changes in substrate association and turnover during partial inactivation of the enzyme (CP6 = 1.0 μM, CP6-K4A = 21 μM and CP6-F6S = 0.2 μM) The average of three independent experiments ($n = 3$) is shown (error bars show the standard error). Source data are provided as Source Data file.

L8K-Ser. This causes ADO-D206 to adopt the same orientation as the peptide-free cobalt-incorporated structure (PDB:8UAN[34], suggesting that substrate (analogue) binding disassembles the hydrogen bonding network promoted by ADO-E92 in the presence of CP6-F6 by enhancing an interaction with the metal centre. Nt-Ser binding is further supported by three protein-mediated interactions: a hydrogen bond between the hydroxyl group of Nt-Ser and the hydroxyl group of Y212, a hydrogen bond between the amine group of Nt-Ser and the carboxylic acid/carboxylate group of D206, and pi-stacking between the amide of Nt-Ser and the sidechain of F101 (Fig. 4).

Following successful implementation of this strategy in generating a structure of ADO in complex with a substrate analogue, additional moieties were installed onto CP6 to examine interactions with extended pseudo-N-terminal sequences. CP6-L8 was replaced with 2,3-diaminopropionic acid (Dap; d), an amino acid that contains a sidechain amine linked to one carbon, from which the dipeptides Lys-

Ser, Phe-Ser, and Gly-Ser were appended, to mimic the N-termini of known NCO substrates (RGS4/5, IL32, and RAP2 [an NCO target in plants[9,10]], respectively; Fig. 3c). Although each substituent reduced binding affinity relative to CP6, with CP6-L8d-Lys-Ser, CP6-L8d-Phe-Ser, and CP6-L8d-Gly-Ser generating $K_D$ values of 490, 79, and 150 nM, respectively, a crystal structure of cobalt-incorporated ADO in complex with CP6-L8d-Gly-Ser was obtained at 1.74 Å resolution (PDB:9DXB; Supplementary Fig. 16, Supplementary Table 9). The pseudo-Nt-Ser residue remains coordinated to the metal centre in a bidentate arrangement, supported by interactions with ADO-F101, -D206 and -Y212, with one water molecule trans to ADO-H112, analogous to CP6-L8K-Ser. No additional interactions are observed except, possibly, weak water-mediated hydrogen bonds between the second amide of CP6-L8d-Gly-Ser and the sidechain hydroxyls of ADO-Y212 and -Y222, as these water molecules are absent in the CP6-L8K-Ser structure.

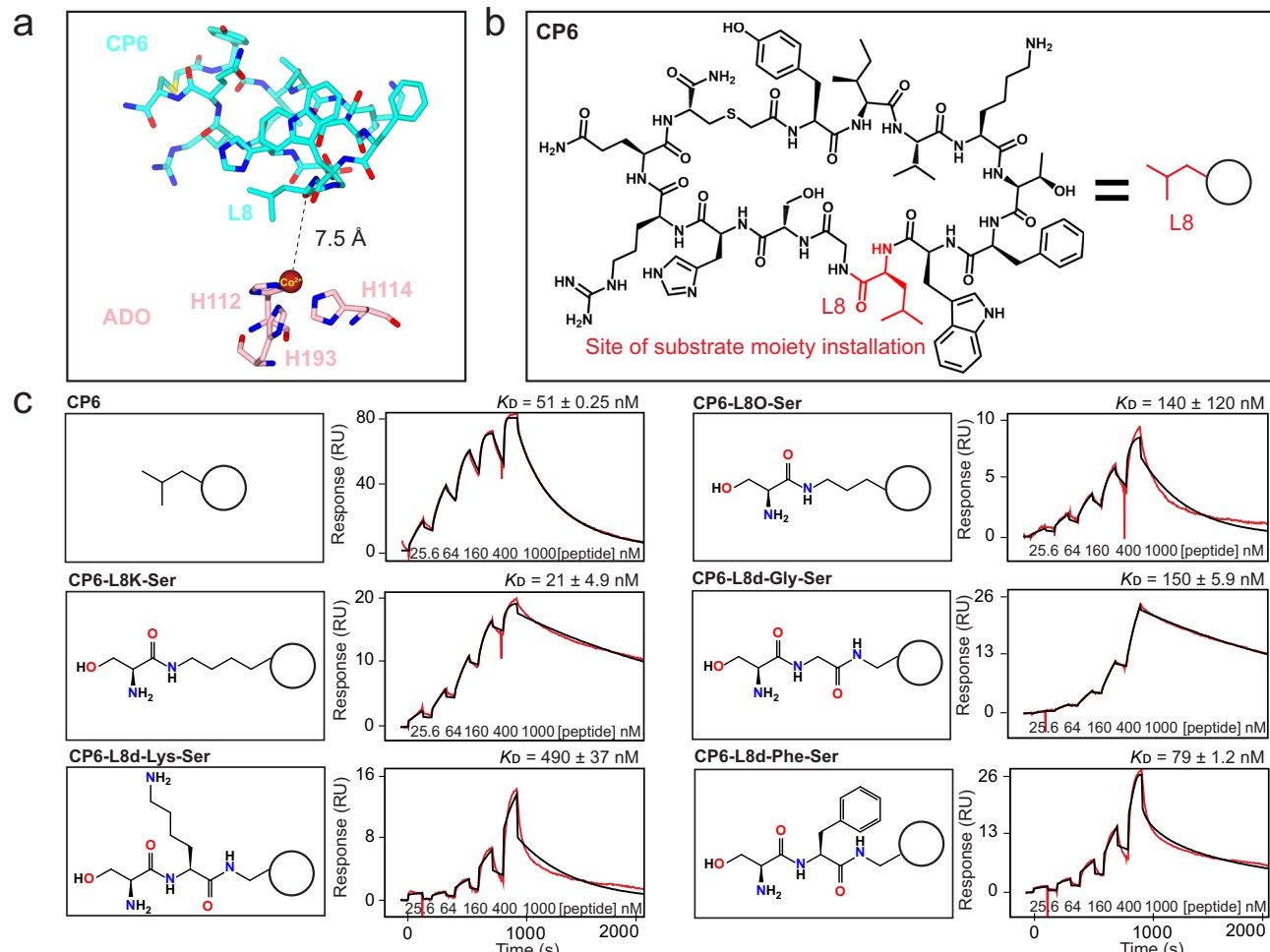

**Fig. 3 | Employing CP6 as a scaffold to graft substrate moieties. a** The crystal structure of cobalt-incorporated ADO (pink) in complex with CP6 (cyan), displayed as cylinders, highlighting the position and distance of CP6-L8 relative to the metal centre of ADO. **b** The chemical structure of CP6, highlighting the position of CP6-L8, the site of substrate moiety installation. **c** Biophysical analysis of CP6 substituents bearing a substrate moiety. (*Left*) Representative chemical structures of the substrate moieties grafted onto CP6 (represented as a circle) through substitutions in CP6-L8. 'O' corresponds to ornithine and 'd' corresponds to 2,3-diaminopropionic acid. (*Right*) SCK SPR sensorgrams of CP6 substituents bearing a substrate moiety with ADO. The sensorgrams are shown in red and the fits to the data are shown in black. The concentrations of CPs used in the titration and equilibrium dissociation constants ($K_D$) are shown ($K_D$ given as the geometric mean of a minimum of three independent SPR measurements ($n = 3$)). Source data are provided as Source Data file.

Together these crystal structures provide molecular descriptions of an ADO substrate complex, highlighting key binding interactions (including bidentate coordination of the Nt-residue), which can inform both reaction mechanism and rational inhibitor design.

## Verifying active site interactions and residues

The crystal structures of cobalt-incorporated ADO in complex with CP6-L8K-Ser and CP6-L8d-Gly-Ser provide valuable information on substrate analogue binding. However, they do not reflect native conditions in which an iron cofactor interacts with an Nt-Cys residue. To corroborate that the crystal structures of cobalt-incorporated ADO in complex with CP6-L8K-Ser and CP6-L8d-Gly-Ser are biologically relevant, molecular dynamic (MD) simulations were conducted with different combinations of active site metal and pseudo-N-terminal substrate residues over 100 ns (Supplementary Fig. 18). In each instance, no changes in hydrogen bonding distance were observed, with Ser and Cys interacting with cobalt(II) and iron(II) through ligation of both their amine and hydroxyl/thiol groups, supporting bidentate coordination of the substrate. The water molecule trans to ADO-H112 remains within hydrogen bonding distance of the metal, with minimal changes in ADO structure, as determined by RMSD (Supplementary Fig. 19a). MD was also used to calculate the binding energies associated

with CP6-L8d-Gly-Ser/Cys complex formation, verifying key residue interactions, most notably the contributions of ADO-Y87 and CP6-W7, ADO-W257 and CP6-C14, ADO-Y212 and CP6-L8d-Gly-Ser/Cys, all of which were highlighted above during assessment of the crystal structure (Supplementary Fig. 19). It also emphasised the role of ADO-Q252, which forms hydrogen bonds with the backbone nitrogen and oxygen of CP6-R12 (Supplementary Fig. 8).

Having verified the accuracy of the substrate analogue-bound crystal structures using MD simulations, key active site residues were mutated to assess their impact on binding and activity using SPR and LCMS. They were chosen based on their proximity to the substrate moieties and their conservation across species, including plants, where NCOs are called plant cysteine oxidases (PCOs; Fig. 5)[9,10]. Two representative substrates, RGS5$_{2-15}$ and IL32$_{2-15}$ (a synthetic peptide corresponding to the methionine excised N-terminus of IL32, residues 2–15), were tested to compare interactions with different amino acid sequences.

From a binding perspective, changes in ADO-F101 and -Y212 caused the largest deviation in dissociation for both substrates, increasing $K_D$ up to 80 and 113-fold, respectively (Fig. 5c *upper panel*, Supplementary Figs. 20 and 21, Supplementary Table 7). This correlates with observations from the crystal structure that they support

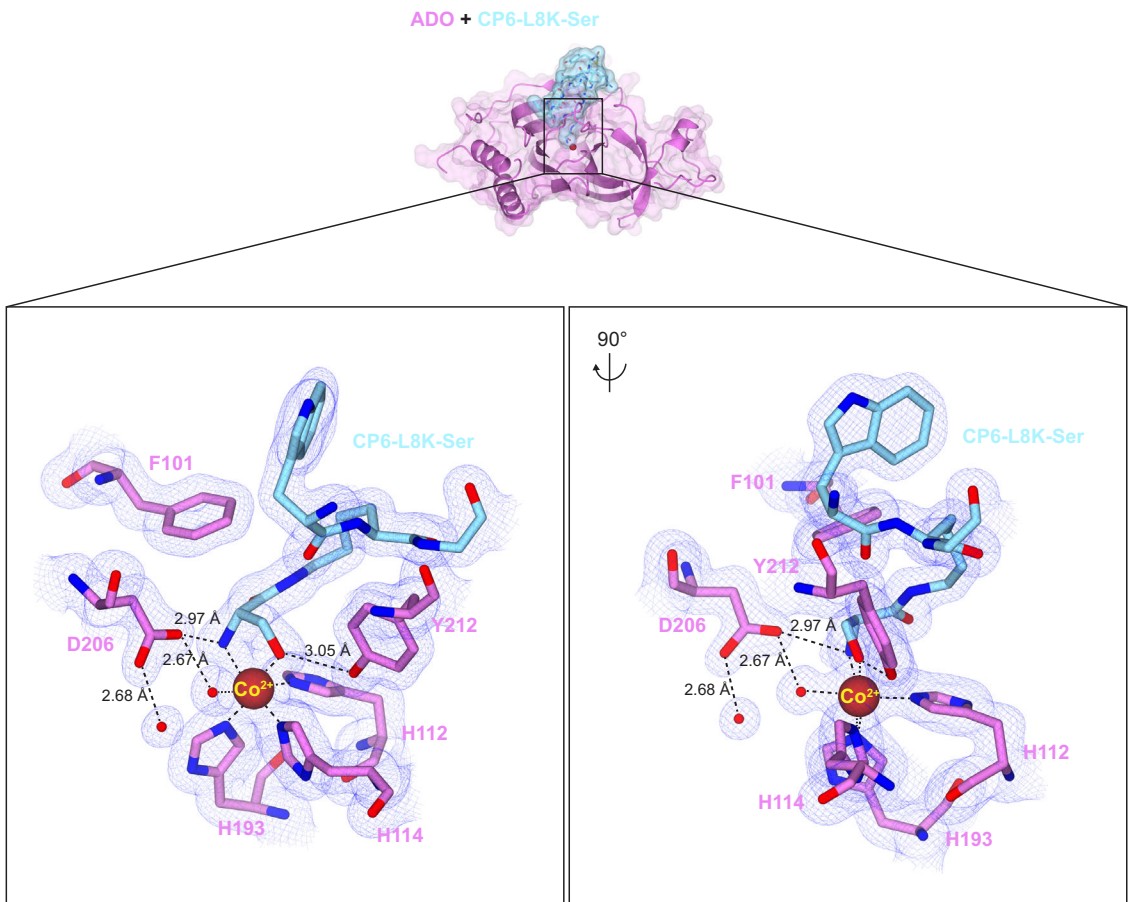

**Fig. 4 | A crystal structure of ADO in complex with CP6-L8K-Ser reveals key substrate binding interactions.** The crystal structure of cobalt-incorporated ADO (dark pink) in complex with CP6-L8K-Ser (light blue; 1.60 Å resolution), displayed through ribbon, surface and cylinder representations, with 2Fo-Fc electron density shown as a blue mesh contoured to 1 sigma. The pseudo-Nt-Ser residue coordinates the metal cofactor in a bidentate arrangement through ligation of the hydroxyl and amine groups (boxed insert). ADO-F101 forms a pi-stacking interaction with the substrate amide, while ADO-Y212 and -D206 form hydrogen bonds with the hydroxyl and amine groups of Nt-Ser, respectively. ADO-D206 also interacts with the water molecule trans to ADO-H112, which is the putative $O_2$ binding site.

Nt-residuebinding through pi-stacking and hydrogen bonding interactions. No binding was observed for ADO-D206A, while ADO-D206N moderately increased affinity 4 to 7-fold. ADO-I109A was the only variant to exhibit a substrate specific change, increasing the $K_D$ of RGS5$_{2-15}$ and IL32$_{2-15}$ 2 and 35-fold, respectively, suggesting it participates in a specific interaction with the latter, possibly residue IL32$_{2-15}$-F3, which is important for binding[36].

Specific activity was measured under saturating substrate concentrations (1 mM) to identify residues involved in catalysis (Fig. 5c *bottom panel*, Supplementary Table 8). While fluctuations in activity were observed, which might reflect changes in substrate binding affinity, particularly for ADO-F101 which exhibited reduced activity with both substrates, mutants of D206 significantly impacted turnover. ADO-D206A and -D206N reduced activity by 99 and 98%, respectively (the latter of which can be attributed to impaired substrate binding), while D206E reduced activity by 80%. This indicates that ADO-D206 is an essential catalytic residue, which facilitates turnover through its carboxylic acid/carboxylate group, as ADO-D206E retains some activity, suggesting ADO-D206 contributes to turnover through acid/base chemistry or its charge properties.

ADO cannot bind oxidised peptide or protein sequences (i.e. those bearing Nt-Cys sulfinic acid/sulfinate)[36]. To explore whether ADO-D206 plays a role in product release through protonation or charge repulsion, the binding of oxidised RGS5$_{2-15}$ and IL32$_{2-15}$ to ADO-D206N was monitored by SPR (Supplementary Fig. 22). No association was observed up to 100 μM, suggesting that ADO-D206 does not promote Nt-Cys sulfinic acid/sulfinate release. Given its position in the active site (and the fact that many ADO-D206 variants bind substrate), this implies that ADO-D206 contributes to ADO activity through an interaction with $O_2$ during turnover.

## Discussion
ADO has emerged as a potential therapeutic target due to its role in $O_2$ sensing and hypoxic adaptation. However, there is no structural information on substrate binding, impeding molecular discovery and enzyme characterisation. To address these issues, we employed a unique strategy by which substrate moieties were grafted onto an mRNA-display derived cyclic peptide scaffold. This allowed the crystal structures of an ADO complex to be determined, which highlights key substrate binding interactions and potential reaction mechanisms. In both structures, the substrate analogue coordinates the metal cofactor in a bidentate arrangement through ligation of both the hydroxyl and amine group, leaving one ligation site available for $O_2$ to bind in an end on orientation, analogous to the homologue CDO[27–29]. This suggests that oxygen activation proceeds through an iron(III) superoxo intermediate, which reacts with the thiol via radical recombination as proposed for other TDOs[30,31]. Whether this occurs through addition of the proximal oxygen in a concerted mechanism, resulting in the formation of a persulfenate intermediate, or through addition of the distal oxygen in a step wise mechanism, which produces an iron(IV) oxo species, remains to be resolved (Supplementary Fig. 23). However, a recent spectroscopic

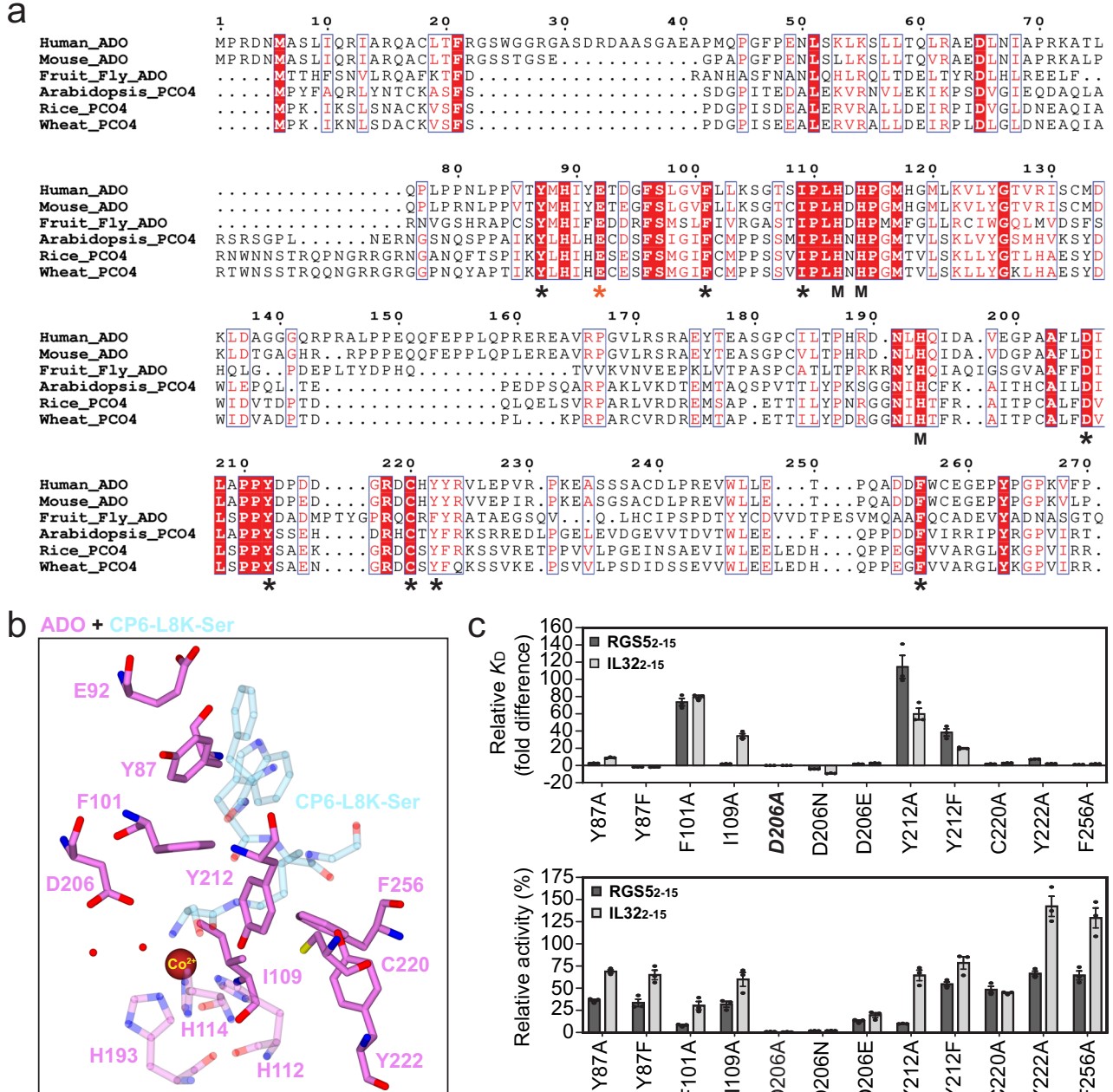

**Fig. 5 | The identification of ADO residues involved in substrate binding and turnover. a** A sequence alignment comparing the amino acid composition of different NCOs, including ADOs (human, mouse and fruit fly) and PCOs (arabidopsis, rice and wheat). Red background and white text highlights conserved residues. M denotes residues involved in metal cofactor coordination. An asterisk (*) denotes residues mutated during this study. A black asterisk denotes residues mutated for substrate binding and activity measurements. A red asterisk (*) denotes ADO-E92, which was mutated for CP6 characterisation. **b** The active site of cobalt-incorporated ADO (dark pink) in complex with CP-L8K-Ser (light blue), highlighting the residues mutated in this study (opaque). **c** Biophysical and activity analysis of ADO mutants with RGS5$_{2-15}$ (black) or IL32$_{2-15}$ (grey). (*Top panel*) The fold difference in equilibrium dissociation constant ($K_D$) relative to wild type ADO ($K_D$ calculated as the geometric mean of a minimum of three independent SPR measurements ($n = 3$); error bars show the standard error). D206A (bold italics) displayed no binding. (*Bottom*) The specific activity relative to wild type ADO. Conducted at 37 °C under aerobic conditions using 1 mM substrate. The average of three independent experiments ($n = 3$) is shown (error bars show the standard error). Source data are provided as Source Data file.

paper provides evidence for the former, as no high valent (i.e. iron(IV) oxo) intermediate was detected in a catalytically impaired ADO complex, which turns over slowly[34].

Despite multiple attempts, we were unable to generate crystals of ADO in complex with CP6-L8K-Cys. This may be the result of thiol oxidisation and/or peptide dimerisation as ADO does not bind sulfinic acid/sulfinate and Nt-Cys readily forms disulphide bonds[11,36]. Nevertheless, MD simulations suggest that the binding mode observed in our crystal structures reflects the native interaction as no change in

hydrogen bonding distance is observed when ferrous iron ($Fe^{2+}$) and Nt-Cys (the native constituents) are added to the model, with both amine and thiol groups remaining in contact with the metal over a 100 ns simulation. Furthermore, previous biophysical experiments have demonstrated that a free N-terminal thiol, a free N-terminal amine, and an active site metal ($Fe^{2+}$, $Zn^{2+}$ or $Co^{2+}$) are vital for ADO substrate engagement, supporting bidentate coordination[35,36]. This work also demonstrated that Nt-Ser can substitute Nt-Cys in terms of binding, albeit at the cost of affinity, rendering it an amenable

analogue for structural studies. Some spectroscopic experiments have proposed that the substrate thiol ligates the iron cofactor in a monodentate arrangement, leaving two coordination sites for $O_2$ to bind in a side-on orientation[32,33]. However, most of these studies used cysteamine instead of an amino acid sequence bearing an Nt-Cys. This renders our structural information more reliable in regard to initial protein substrate complex formation. Nevertheless, we cannot rule out that changes in metal coordination occur during turnover, as observed for other non-haem iron-dependent enzymes[44–46].

NCOs have low sequence and structural homology with CDO, which processes free L-Cys as part of sulphur metabolism. However, ADO has a distinct distribution of amino acids in its active site (Supplementary Fig. 24)[24,25]. While mutagenesis revealed that many of these differences facilitate the binding of specific substrate, such as ADO-F101, which forms a pi-stacking interaction with the substrate amide, ADO-D206 was identified as a unique catalytic residue. Activity assays with various ADO-D206 mutants revealed that the carboxylate/carboxylic acid group is important for turnover. However, biophysical measurements indicated that it does not contribute to substrate binding (by deprotonating the Nt-thiol or -amine) or product release. Instead, ADO-D206, which sits within hydrogen bonding distance of the putative $O_2$ coordination site, may orientate and/or stabilise a reactive oxygen intermediate, ensuring correct reaction with the substrate thiol. A similar role has been proposed for residue Y157 in CDO (rat numbering)[47,48]. Given its position in the active site (it sits above the Nt-residue of the substrate, facing the same direction as the Nt-hydroxyl and Nt-amine; Fig. 4), ADO-D206 may orientate the distal oxygen of the putative iron(III) superoxo species away from the thiol group through charge repulsion or orbital overlap, promoting initial reaction with the proximal oxygen (Supplementary Fig. 25a). Alternatively, ADO-D206 may direct the putative iron(III) superoxo species by transiently protonating the distal oxygen, increasing the electronegativity and reactivity of the proximal oxygen (Supplementary Fig. 25b), stabilise the reaction by preventing protonation of a reactive oxygen intermediate (Supplementary Fig. 26a), or maintain the protonation state of the water molecule occupying the $O_2$ binding site (Supplementary Fig. 26b). The latter would reduce hydroxide formation, facilitating exchange with $O_2$. A similar role has been proposed for Y159 in the TDO 3-mercaptopropionate dioxygenase[49]. Another difference between ADO and CDO is substrate orientation. The substrate amine and (would be) thiol groups occupy opposite coordination sites on the metal cofactor (Supplementary Fig. 24)[27–29]. Unlike other eukaryotes, mammals only encode one NCO[7,10]. While not observed in our crystal structures, it is possible that ADO binds its substrates in both orientations to increase promiscuity. This would allow the enzyme to regulate the stability of different proteins, consisting of different amino acid sequences, through the N-degron pathway, as suggested by activity assays[7,35,36]. This may explain variations in binding interactions observed during this investigation (Figs. 3b, and 4c). However, additional mutagenesis, substrate binding and structural analysis is required to verify this claim.

This work also established selective modulators of ADO, a potential target for hypoxic disease. The CPs were shown to inhibit ADO through different mechanisms. CP6 impacted substrate turnover, as well as binding, through an indirect interaction with D206, emphasising the role of this residue in catalysis. While it is unlikely that the CP inhibitors described in this report will be of immediate pharmaceutical value, they do highlight important interactions which, in conjunction with the substrate analogue structures, can be used to rationally design more drug-like inhibitors. Many of the residues examined in this report are conserved in other NCOs, including the PCOs (Fig. 5A), which regulate the stability of transcription factors linked to crop flood resistance[9,10,43,50–52]. As a result, these structures can also inform the design and optimisation of inhibitors with agrichemical applications. Accordingly, in addition to elucidating enzyme mechanism, this work

will facilitate the development of new chemical candidates to study and manipulate the Cys branch of the N-degron pathway and associated low $O_2$ processes, which, in turn, may lead to new strategies to address hypoxic disease and improve crop stress tolerance.

## Methods

### Protein production

Full-length human ADO (Uniprot ID: Q96SZ5) was cloned into the pET28a (thrombin cleavage site following tag) and pETDuet (TEV cleavage site following tag) plasmids for bacterial expression as N-terminal His-tagged proteins and into the pQE80L-Navi plasmid for bacterial expression as an N-terminally His-tagged and biotinylated protein. The pET28a-ADO construct was used to produce protein for enzyme kinetic assays, the pETDuet-ADO construct was used to produce protein for X-ray crystallography, and the pQE80L-Navi-ADO construct was used to produce protein for SPR experiments. Full-length human RGS5 (Uniprot ID: O15539) was cloned into pETDuet with an N-terminal His- and SUMO-tag for bacterial expression. The pETDuet-His-SUMO-RGS5 construct was used to produce all RGS5 proteins used in this study. Site-directed mutagenesis was used to introduce mutations in the constructs listed above.

All constructs were transformed into Rosetta2(DE3) *Escherichia coli* (*E. coli*) cells in preparation for protein expression. 2xYT, supplemented with the appropriate antibiotics, was used as the expression medium to express standard proteins. Expression cultures were inoculated with saturated overnight cultures (1:100 dilution) prepared using single colonies from fresh transformations or glycerol stocks. Expression cultures were incubated at 37 °C, with shaking at 120–150 rpm, and allowed to grow to an OD600 of ~0.6–0.8 before being cooled to room temperature and supplemented with 0.5 mM IPTG (and 0.2 mM biotin for the pQE80L-Navi construct to enable biotinylation) to induce expression. Expression cultures were further incubated at 20 °C, with shaking, for ~18–24 h before harvesting via centrifugation at 4000 $g$ for 25 minutes. Cell pellets were stored at −20 °C until required for protein purification.

Cobalt-incorporated proteins were produced as described above for standard proteins up until OD600 of ~0.6–0.8 was reached. At this point, cultures were harvested (15 minutes at 4000 $g$) and cell pellets were resuspended in M9 minimal media supplemented with 0.2 mM $CoCl_2$. A volume of minimal media half the volume of the original 2xYT expression culture was used for resuspension and the appropriate antibiotics were added. The cultures were then transferred to 20 °C, with shaking, for 45 minutes before expression was induced with 0.5 mM IPTG and allowed to proceed as described above.

### Protein purification

**ADO prepared for RaPID screening and SPR experiments (pQE80L-NAvi-ADO).** A purification procedure of nickel-ion ($Ni^{2+}$) affinity chromatography followed by size exclusion chromatography was used to prepare the ADO used for SPR in this study. Cell pellets were resuspended in lysis buffer (20 mM HEPES pH 7.5, 500 mM NaCl, 1 mM TCEP, 20 mM imidazole, 10 µg/ml DNAse I, 100 µg/ml lysozyme, and 1x cOmplete EDTA-free protease inhibitor) and lysed by sonication before being clarified via centrifugation (17,000 g for 30 minutes followed by filtration through a 0.45 µM membrane). The soluble fractions of cell lysates were subject to $Ni^{2+}$-affinity chromatography by incubating the supernatant with Ni-NTA agarose (Cytiva) equilibrated in wash buffer (20 mM HEPES pH 7.5, 500 mM NaCl, 1 mM TCEP, and 20 mM imidazole). Bound proteins were washed and eluted with elution buffer (20 mM HEPES pH 7.5, 500 mM NaCl, 1 mM TCEP, and 500 mM imidazole). Protein-containing fractions were concentrated and injected onto an equilibrated 26/600 HiLoad Superdex 75 prep grade column (Cytiva) and eluted using SEC buffer (20 mM HEPES pH 7.5, 150 mM NaCl, and 1 mM TCEP). Protein purity was assessed by SDS-PAGE and concentrations were determined using A280 nm measurements.

**ADO prepared for enzyme assays (pET28a-ADO).** A purification procedure of nickel-ion (Ni$^{2+}$) affinity chromatography followed by size exclusion chromatography was used to prepare the ADO used for enzyme assays in this study. Cell pellets were resuspended in lysis buffer (50 mM Tris pH 7.5, 400 mM NaCl, 20 mM imidazole, 10 μg/ml DNAse I, and 1x cOmplete EDTA-free protease inhibitor) and lysed by sonication before being clarified via centrifugation (17,000 g for 30 minutes followed by filtration through a 0.45 μM membrane). The soluble fractions of cell lysates were subject to Ni$^{2+}$-affinity chromatography by passing the supernatant through a HisTrap HP column (Cytiva) equilibrated in wash buffer (50 mM Tris pH 7.5, 400 mM NaCl, 20 mM imidazole). Bound proteins were washed and eluted using a 20 mM to 1 M imidazole gradient in a base buffer comprised of 50 mM Tris pH 7.5 and 400 mM NaCl. Protein-containing fractions were concentrated and buffer exchanged into SEC buffer (50 mM Tris pH 7.5 and 400 mM NaCl) using a PD-10 desalting column (Cytiva). The desalted protein was injected onto an equilibrated 26/600 HiLoad Superdex 75 prep grade column (Cytiva) and eluted using SEC buffer. Protein purity was assessed by SDS-PAGE and concentrations were determined using A280 nm measurements.

**ADO prepared for X-ray crystallography (pETDuet-TEV-ADO).** A purification procedure of cobalt-ion (Co$^{2+}$) affinity chromatography, followed by TEV cleavage for His-tag removal, and size exclusion chromatography was used to prepare the ADO used for X-ray crystallography in this study. Cell pellets were resuspended in lysis buffer (20 mM HEPES pH 7.5, 500 mM NaCl, 0.1 μM CoCl$_2$, 2.5 mM imidazole, 10 μg/ml DNAse I, 100 μg/ml lysozyme, and 1x cOmplete EDTA-free protease inhibitor) and lysed by sonication before being clarified via centrifugation (17,000 g for 30 minutes followed by filtration through a 0.45 μM membrane). The soluble fractions of cell lysates were subject to Co$^{2+}$-affinity chromatography, using Talon resin (Cytiva) equilibrated in wash buffer (20 mM HEPES pH 7.5, 500 mM NaCl, 0.1 μM CoCl$_2$, and 2.5 mM imidazole). Bound proteins were washed and eluted with elution buffer (20 mM HEPES pH 7.5, 500 mM NaCl, 1 mM TCEP, and 300 mM imidazole). Eluates were concurrently dialysed in SEC buffer (20 mM HEPES pH 7.5, 150 mM NaCl, and 1 mM TCEP) and incubated with TEV protease overnight at 4 °C to cleave the His-tag. The dialysed cleaved protein was passed through Talon resin (Cytiva) for His-tag removal before being concentrated and injected onto an equilibrated 16/600 HiLoad Superdex 75 SEC column (Cytiva). Protein was eluted from the column using SEC buffer. Protein purity was assessed by SDS-PAGE and concentrations were determined using A280 nm measurements.

**RGS5 prepared for SPR competitive experiments (pETDuet-RGS5).** A purification procedure consisting of nickel-ion (Ni$^{2+}$) affinity chromatography, followed by Ulp1 cleavage for His-SUMO-tag removal, and size exclusion chromatography was used to prepare RGS5. Cell pellets were resuspended in lysis buffer (20 mM HEPES pH 7.5, 500 mM NaCl, 1 mM TCEP, 20 mM imidazole, 10 μg/ml DNAse I, 100 μg/ml lysozyme, and 1x cOmplete EDTA-free protease inhibitor) and lysed by sonication before being clarified via centrifugation (17,000 g for 30 minutes followed by filtration through a 0.45 μM membrane). The soluble fractions of cell lysates were subject to Ni$^{2+}$-affinity chromatography by incubating the supernatant with Ni-NTA agarose (Cytiva) equilibrated in wash buffer (20 mM HEPES pH 7.5, 500 mM NaCl, 1 mM TCEP, and 20 mM imidazole). Bound proteins were washed and eluted with elution buffer (20 mM HEPES pH 7.5, 500 mM NaCl, 1 mM TCEP, and 500 mM imidazole). Eluates were concurrently dialysed in SEC buffer (20 mM HEPES pH 7.5, 150 mM NaCl, and 1 mM TCEP) and incubated with Ulp1 protease overnight at 4 °C to cleave the His-SUMO-tag. The dialysed cleaved protein was passed through Talon resin (Cytiva) for His-SUMO-tag removal before being concentrated and injected onto an equilibrated 16/600 HiLoad Superdex 75 SEC column (Cytiva). Protein was eluted from the column using SEC buffer.

Protein purity was assessed by SDS-PAGE and concentrations were determined using A280 nm measurements.

## Random non-standard peptide integrated discovery (RaPID) mRNA display

RaPID was conducted as previously described[37–40]. Briefly, DNA oligonucleotides comprising a T7 promoter, ribosome binding site, ATG start codon, 4–15 NNS (N = A, C, G or T; S = C or G) codons, a TGC (Cys) codon and a 3' fixed region encoding a Gly-Asn-Leu-Ile linker were amplified by PCR (see Table 2 for specific nucleotide sequences). The resulting DNA libraries were transcribed in vitro using T7 RNA polymerase to generate cognate mRNA libraries, which were purified by denaturing urea polyacrylamide gel electrophoresis. The libraries of different lengths were then pooled proportional to theoretical diversity to generate an RNA library. These were ligated to a puromycin linked oligonucleotide using T4 RNA ligase. Ribosomal synthesis of the macrocyclic peptide library from the puromycin-linked mRNA library was performed using the PURExpress ΔRF123 kit (New England Biolabs) with RF2 and RF3 added. To allow for genetic code reprogramming, a custom "Solution A" was used supplemented with 19 amino acids (-Met) and N-chloroacetyl-L-tyrosine aminoacylated initiator tRNA as previously described[53]. Following translation, the ribosome was denatured by addition of EDTA, and the resulting mRNA-peptide library was reverse transcribed using RNase H- reverse transcriptase.

The reverse transcribed peptide-mRNA libraries were panned against 200 nM biotinylated target protein immobilised on Dynabeads M-280 streptavidin (Life Technologies) for 30 min at 4 °C. After washing with Tris buffered saline (50 mM Tris pH 8.0, 150 mM NaCl) containing 0.05% tween-20 (TBS-T), the fused peptide–mRNA/cDNA was isolated from the beads by heating to 95 °C for 5 min, and cDNA was amplified by PCR, purified by ethanol precipitation and transcribed as above to produce the enriched mRNA library for the next round of selection. For the second and subsequent rounds of selection, three iterative counter-selections were used to remove peptides with affinity for the streptavidin beads. Sequencing of the final enriched cDNA was conducted using an iSeq next generation sequencer (Illumina).

## Fmoc-solid-phase peptide synthesis (SPPS)

Please see supporting information, supplementary methods.

## Surface plasmon resonance

SPR measurements were made using a Biacore T200 (Cytiva) instrument and data were analysed using the Biacore Insight Evaluation Software. Biotinylated-ADO was immobilised on a Biotin CAP chip (Cytiva) with a target density of ~2500–3000 RU and peptide/protein substrates were injected over the chip. Experiments were conducted at 4 °C using both multi-cycle (fit using the equilibrium steady state affinity 1:1 binding model) and single-cycle kinetics mode (fit using a 1:1 binding model). A buffer comprising 20 mM HEPES pH 7.5, 500 mM NaCl, 10 mM DTT, and 0.1% (v/v) tween-20 was used as the running buffer at a flow rate of 50 μL/minute.

## Activity assay

The enzymatic activity of ADO was examined by incubating between 0–1000 μM peptide substrates with 0.1 μM ADO in a bench-top thermomixer at 37 °C for 45 seconds. Otherwise stated, the reaction buffer contained 50 mM Tris pH 7.5, 50 mM NaCl, and 5 mM TCEP. The reactions were quenched by mixing the sample 1:10 with 1% formic acid (v/v).

Peptide samples were injected onto a Chromolith® RP-18 Endcapped HPLC Columns (100–2 mm; Merck), heated to 40 °C and eluted at 0.3 ml/min using a gradient of 95% deionised water supplemented with 0.1% (v/v) formic acid to 95% acetonitrile. Oxidation was monitored by an ultra-high-performance liquid chromatography

**Table 2 | The oligonucleotide and primer sequences used for library generation and selection**

| Name | Sequence (5' to 3') |
|---|---|
| Template 4 | TTAAGAAGGAGATATACATATG(NNS)$_4$TGCGGTAACTTAATCTAGG |
| Template 5 | TTAAGAAGGAGATATACATATG(NNS)$_5$TGCGGTAACTTAATCTAGG |
| Template 6 | TTAAGAAGGAGATATACATATG(NNS)$_6$TGCGGTAACTTAATCTAGG |
| Template 7 | TTAAGAAGGAGATATACATATG(NNS)$_7$TGCGGTAACTTAATCTAGG |
| Template 8 | TTAAGAAGGAGATATACATATG(NNS)$_8$TGCGGTAACTTAATCTAGG |
| Template 9 | TTAAGAAGGAGATATACATATG(NNS)$_9$TGCGGTAACTTAATCTAGG |
| Template 10 | TTAAGAAGGAGATATACATATG(NNS)$_{10}$TGCGGTAACTTAATCTAGG |
| Template 11 | TTAAGAAGGAGATATACATATG(NNS)$_{11}$TGCGGTAACTTAATCTAGG |
| Template 12 | TTAAGAAGGAGATATACATATG(NNS)$_{12}$TGCGGTAACTTAATCTAGG |
| Template 13 | TTAAGAAGGAGATATACATATG(NNS)$_{13}$TGCGGTAACTTAATCTAGG |
| Template 14 | TTAAGAAGGAGATATACATATG(NNS)$_{14}$TGCGGTAACTTAATCTAGG |
| Template 15 | TTAAGAAGGAGATATACATATG(NNS)$_{15}$TGCGGTAACTTAATCTAGG |
| Forward primer | TAATACGACTCACTATAGGGTTAACTTTAAGAAGGAGATATACATA |
| Reverse primer | TTTCCGCCCCCCGTCCTAGATTAAGTTACCGCA |

**Table 3 | A summary of the parameters used during MD simulations**

| System | Ser-Co$^{2+}$ | Ser-Fe$^{2+}$ | Cys-Co$^{2+}$ | Cys-Fe$^{2+}$ |
|---|---|---|---|---|
| Simulation box dimensions (10 Å buffer) | 68.955 Å × 66.128 Å × 90.245 Å | 68.971 Å × 66.143 Å × 90.200 Å | 68.954 Å × 66.129 Å × 90.242 Å | 68.971 Å × 66.188 Å × 90.197 Å |
| Total number of atoms | 38479 | 38428 | 38467 | 38455 |
| Total number of water molecules | 11513 | 11496 | 11509 | 11505 |
| Salt concentration | 0.15 M NaCl | 0.15 M NaCl | 0.15 M NaCl | 0.15 M NaCl |

(UHPLC) mass spectrometry (MS) using ExionLC AD (Sciex) and SelexION (Sciex) mass spectrometer operated in a positive electro-spray time-of-flight (+ESI-TOF) mode. Instrument parameters, data acquisition and data processing were controlled by Analyst software (Sciex). The peptide data processing was performed using Skyline software[54]. Turnover was quantified by comparing the areas underneath the product and substrate ions extracted from the total ion current chromatogram. All figures and parameters were generated using GraphPad Prism 9.

## X-ray crystallography

Crystallisation of ADO-peptide complexes was performed using a sitting-drop vapour-diffusion technique. Low concentrations (~20–50 μM) of purified ADO were combined with ~1 molar equivalent of peptide (dissolved in 50 mM Tris pH 7.5, 400 mM NaCl) and concentrated to the levels required for crystallisation (~2–3 mM). Initial crystallisation trials were performed using commercial 96-well crystallisation screens (JCSG-plus, PEGRx HT and PACT Premier). ADO-peptide mixtures were dispensed into MRC two-drop chamber, 96-well crystallisation plates using a Mosquito crystallisation robot and each condition was screened at a 1:1 or 2:1 protein to precipitant ratio (maintaining a final drop volume of 300 nL). Initial hits were optimised by scaling up drop sizes and microcrystal seeding. All experiments were performed at 18 °C. Protein crystals generally took days to weeks to appear. Crystals were frozen by plunge-freezing in liquid nitrogen following cryoprotection with 10% glycerol in the mother liquid from which the crystals were grown. Final crystallisation conditions were (1) ADO CP6: 20% (w/v) PEG 3350 and 0.2 M Sodium thiocyanate, (2) ADO CP6-L8K-Ser: 20 % (w/v) PEG 3350, 0.2 M Sodium nitrate and 0.1 M Bis-Tris propane pH 6.5, and (3) ADO CP6-L8d-Gly-Ser: 25 % (w/v) PEG 3350 and 0.1 M BIS-Tris pH 5.5.

X-ray diffraction data were collected from frozen crystals at the Australian Synchrotron using the Macromolecular Crystallography MX1 (bending magnet) and MX2 beamlines (microfocus) at 100 K and a wavelength of 0.9537 Å[55,56]. Data were integrated using XDS and were processed further using the CCP4i suite[57,58]. AIMLESS was used for indexing, scaling, and merging of the data and the initial phases were calculated by the molecular replacement programme PhaserMR using existing x-ray structures of ADO as the molecular replacement models (PDB: 7REI)[25]. Manual model building was performed using COOT and refinement was performed by iterative rounds of manual building in COOT followed by refinement using Phenix[59,60]. The quality of the final model was validated with the wwPDB server and submitted to the PDB. Structure diagrams were generated using CCP4MG[61]. The data collection and refinement statistics for all structures described in this study are outlined in Supplementary Table 9.

## Molecular dynamics

Molecular dynamics was utilised to study the coordination bonds between the metal (Co$^{2+}$ or Fe$^{2+}$), ligand (CP6-L8d-Gly-Ser or -Cys) or coordinated water molecule. From the original structure, the Ser to Cys ligand variant was generated using the mutation function in Maestro (Schrödinger Release 2024-2: Maestro, Schrödinger, LLC, New York, NY, 2024). The Co$^{2+}$ metal atom was changed to Fe$^{2+}$ manually. The crystal structures were prepared in Maestro using the Protein Preparation Wizard[62], with all waters not within 5 Å of a heteroatom removed. A total of four simulations were run: cobalt-serine ligand, cobalt -cysteine ligand, iron-serine ligand and iron-cysteine ligand. The aim of the MD simulations to study the coordination bonds between the metal, ligand and coordinated water do not expect to cause major conformation changes without large perturbations. As such, enhanced sampling methods which usually introduce external potentials were not required.

For the molecular dynamics studies, an orthorhombic solvent box filled with SPC water molecules, counter ions and additional 0.15 M concentration of NaCl was constructed in Desmond system builder around each system (Table 3). The protein and ligands were parameterised using the OPLS4 forcefield, which has been shown to have

improved accuracy in assessing protein-ligand binding[63]. Simulations were performed for 100 ns both on each protein complex after standard energy minimisation and equilibration. The simulations were run at 310 K, regulated via the Nosé–Hoover thermostat and standard pressure, regulated via the MTK (Martyna-Tobias-Klein) barostat[64,65]. Simulations were confirmed to have reached equilibrium via analysis of the root mean square deviation (RMSD) of the protein-ligand complex. All simulations were carried out in triplicate. To avoid biased or non-generalisable results caused by the initial configuration, the equilibration and productions runs were conducted with different random seed which assigned different starting Boltzmann distribution of velocities on atoms.

MM-GBSA (Molecular mechanics with generalised Born and surface area solvation) calculations based on the equilibrium trajectory of conventional MD for the interactions between the protein-metal complex and ligand were calculated using Prime in Maestro[66,67]. Every 5th frame from 50 ns to the end of the simulation was taken for the energy calculations.

### Reporting summary

Further information on research design is available in the Nature Portfolio Reporting Summary linked to this article.

## Data availability

The coordinates of cobalt-incorporated ADO in complex with CP6 were deposited to the Protein Data Bank (PDB) under accession code 9DXU. The coordinates of cobalt-incorporated ADO in complex with CP6-L8K-Ser were deposited to the PDB under accession code 9DXV. The coordinates of cobalt-incorporated ADO in complex with CP6-L8d-Gly-Ser were deposited to the PDB under accession code 9DXB. The coordinates of cobalt-incorporated ADO in the absence of a CP were obtained from the PDB using accession code 8UAN. The coordinates of iron-incorporated CDO in complex with Cys were obtained from the PDB using accession code 4IEV. Source data are provided with this paper.

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

## Acknowledgements

We acknowledge Sydney Analytical Core Facilities (University of Sydney) for providing access to SPR infrastructure and Sydney Mass Spectrometry (University of Sydney) for providing access to mass spectrometry infrastructure. We thank Dr Lorna Wilkinson-White and Dr Atul Bhatnager for consultation regarding SPR and mass spectrometry experiments, respectively. We thank the Structural Biology Group, School of Life and Environmental Sciences (University of Sydney), for providing access to protein production and characterisation infrastructure. We thank Prof Joel P. Mackay and Dr Leo Corcilius for discussions on the work. This research was undertaken using the MX1 and MX2 beamlines at the Australian Synchrotron, part of Australian Nuclear Science and Technology Organisation (ANSTO) and made use of the Australian Cancer Research Foundation (ACRF) detector. This work was supported by an Australian Research Council (ARC) Discovery Early Career Researcher Award (DECRA, project ID: DE190100668), a National Health and Medical Research Council (NHMRC) Investigator Grant (project ID: 2008546), and a New South Wales (NSW) Cardiovascular Research Capacity Programme Grant (project ID: EMC72) awarded to M. D. W, and the Australian Research Council Centre of Excellence for Innovations in Peptide and Protein Science (project ID: CE200100012) awarded to R. J. P.

## Author contributions

M.D.W. conceived and designed the study with input from all authors. Y.J., K.P., J.J-L., J.W.C.M., Y.C., J.J.D., and T.P. conducted the

experimental investigation. Y.J. and K.P. performed biochemical and structural analysis. J.J.-L. and J.W.C.M. performed peptide synthesis. Y.C. and J.J.D. performed MD simulations. T.P. performed mRNA display. All authors analysed data and generated figures. M.D.W. and R.J.P. acquired funding. M.D.W., R.J.P., K.M.C., and J.J.D. provided supervision. M.D.W. wrote the manuscript with input from all authors.

## Competing interests

The authors declare no competing interests.
