## [Transparent Peer Review file · Nature Communications]

An mRNA-display derived cyclic peptide scaffold reveals the substrate binding interactions of an N-terminal cysteine oxidase

Corresponding Author: Dr Mark White

Version 0:

Reviewer comments:

Reviewer #1

(Remarks to the Author)

N-degron pathway dictates the half-life of a protein based on N-terminal amino acid residues, some of which are modified from translated side chains. Especially, N-terminal Cys is a tertiary destabilizing residue in the Arg/N-degron pathway and could be oxidized to secondary destabilizing residue by N-terminal cysteine oxidase (NCO), which enhances arginylation for further protein degradation processes. One of major problems in the Arg/N-degron pathway is that we do not know physiological protein substrates of enzymes that convert tertiary destabilizing residues to secondary destabilizing residues (NCO, NTAN1, and NTAQ1). Thus, this manuscript is of great importance in that the authors tried to find substrates of NCO using an mRNA-display derived cyclic peptide scaffold (RaPID)

Firstly, the authors identified and characterized cyclic peptide (CP) inhibitors of the NCO 2-aminoethanethiol dioxygenase (ADO) using RaPID. Eight most enriched CPs (CP1-8) were selected, produced by solid-phase peptide synthesis, and analyzed by in vitro interaction assay SPR using single cycle kinetic analysis. Interestingly, the authors could determine the crystal structure of ADO in complex with CP6 after careful comparison with other crystal structures of unbound ADO. With the CP6-bound ADO structure in hands, the authors could elucidate the substrate interacting modes and provide reasonable reaction mechanism models. In addition, CP6 was used as a scaffold to graft substrate moieties and some substrate analogues, including CP6-L8K-Ser and CP6-L8d-Gly-Ser, were selected for further kinetic analysis and structure determination. The first crystal structure of an ADO substrate complex could provide valuable structural information at an atomic level about reaction mechanism and rational inhibitor design.

Even though metal substituted ADO and substrate analogue residues were used, the authors tried their best to connect their findings to native/physiological interaction using currently available methods, such as comprehensive mutational studies and MD simulation. Overall, this manuscript is very well written and the experimental description in this manuscript was sufficiently and scientifically sound. This manuscript could be published in the journal with very minor revision as described in the below.

1. Introduction: The ADO reaction mechanism is described as texts very in detail. Even this reviewer could not follow completely without visual supports. Thus, please add a schematic diagram/figure about the reaction mechanism that the authors describe.

2. Discussion: The fourth paragraph (Another difference ~~~~ this claim.) seems to be somewhat abrupt and to focus on a specific aspect of their findings with speculation and repetitive/iterative description with the Results section. This fourth paragraph could be shortened and might be connected to the third paragraph in the Discussion section. Instead, the authors could discuss possible extension of their findings to the N-degron pathway, followed by the current discussion about its disease/agricultural relation (the fifth paragraph).

3. Typo in Abstract: 2-amino(e)thanethiol dioxygenase (ADO).

Reviewer #2

(Remarks to the Author)

Jiramongkol et. al describe the determination of the ADO crystal structure in complex with pseudo-substrate analogues. No crystal structure of ADO with substrates has been reported prior to this work. This is attributed to the rapid substrate turnover and low substrate affinity. The authors took advantage of the RaPID mRNA display approach to find cyclic peptide binders of ADO, which were then used as tools to graft substrate analogues of ADO bearing a pseudo-Nt-Cys or -Ser residue. This aided stabilisation of the substrate analogues and led to the determination of two crystal structures of ADO with different pseudo-substrates. This structural data provide evidence on the binding mode of ADO substrates, strongly pointing to a bidentate association of the substrate which can inform both reaction mechanism and inhibitor design. Further structure-guided mutagenesis experiments highlight the importance of D206 in the catalytic mechanism of ADO. Overall, this manuscript provides valuable structural information on the ADO substrate binding mechanism, including strong evidence for a bidentate binding mode – something that has been disputed to this point. More generally it also highlights the use of de novo cyclic peptides as substrate-grafting tools. On the whole the data is sound and well presented and the manuscript is very well written.

On the basis of this I anticipate it will be of great interest to a broad range of Nature Communication readers and recommend acceptance once the authors address the following minor issues:

- More experimental details on the RaPID selection process should be added to Supporting Information. For example, full DNA library sequences used for selection, comparison of enrichment for positive and negative selections, next-generation sequencing data.
- Define the competitive vs. uncompetitive inhibitor nomenclature used in the manuscript – by definition, an uncompetitive inhibitor binds to the enzyme-substrate complex, enhancing the K_m for the substrate whilst inhibiting activity but the competitive SPR e.g. in SF4 shows peptides still block substrate binding (in fact the “uncompetitive” peptide CP6 seems to be one of the most effective substrate competitors) and the crystal structure shows the peptide binding where the substrate has to bind suggesting it is unlikely the peptide can bind to the enzyme-substrate complex. Please expand on this.
- Page 4 - provide a reference for the ADO activity assay used since it is an “established stopped assay”.
- Fig. 1D, Table 1, Supplementary Figures 3 and 9 and Supplementary Table 4 - provide errors in the IC50 values obtained.
- Page 5 and 7 – “root mean squared deviation (rmsd)” – RMSD should be in capital letters.
- Page 6 – “In the presence of ADO-E92A, CP6 exhibited characteristics that more closely align with a competitive inhibitor, increasing the apparent K_m and k_{cat} relative to the DMSO control and wild-type ADO, respectively.” Based on Supplementary Figure 10, the k_{cat} decreases.
- Page 6 – the authors’ produce two substrate analogues of CP6, CP6-L8K-Ser and CP6-L8K-Cys. They crystallised ADO in complex with CP6-L8K-Ser at 1.60 Å resolution, even though CP6-L8K-Cys appears to be a more physiologically-relevant substrate. Has the CP6-L8K-Cys analogue been challenging to crystallise? Please elaborate on this. Can the authors test whether the Cys variants are synthetic substrates for ADO?
- Page 8 - typo “moities” – moieties
- Page 12 – space missing between mRNA and libraries
- Page 12 – “To allow for genetic code reprogramming, a custom “Solution A” was used supplemented with 19 amino acids (-Met) and N-chloroacetyl-L-tyrosine aminoacylated initiator tRNA as previously described [REFS].” References missing.
- Page 12 – “After washing, the fused peptide–mRNA/cDNA was isolated from the beads by heating to 95 °C for 5 min, and cDNA was amplified by PCR, purified by ethanol precipitation and transcribed as above to produce the enriched mRNA library for the next round of selection.” What was the fused peptide–mRNA/cDNA washed with specifically?
- Page 13 – which commercial crystallisation screens were used?
- Page 14 – “Ribosomal synthesis of the macrocyclic peptide library from the puromycin-linked mRNA library was performed using the PURExpress DRF kit (New England Biolabs) with RF2 and RF3 added.” Define D in DRF kit – delta?
- Provide pub ID of the crystal structures obtained.
- The authors expressed and purified several ADO mutants, however SDS-PAGE gels are missing. They should be added to the Supporting Information along with the sequence of the parent ADO construct.
- Fig. 2C – include concentration of peptide
- Fig. 2C – the data on CP6-F6S is confusing. CP6-F6S is more potent (by IC50 and KD – SF9) than CP6 but the Michaelis-Menten kinetic plots show that it doesn’t alter k_{cat} or K_m ? Please can the authors elaborate on this.
- Fig. 3C – I assume that d is 2,4-diaminobutyric acid? Please define in the figure legend
- Fig. 3 (legend) – typos; “ADO-206 also interacts with the water molecule trans to ADO-His112, which is the putative O2 binding site.” Change to ADO-D206 and ADO-H112 (for consistency).
- Supplementary Fig. 2 – The derived KD of CP2 and CP3 is 5-7x higher than the highest concentration of CP2 used in the experiment. Can the authors comment on how accurate they believe this KD value? Also, can they comment on the accuracy of the CP8 fit which appears to have mostly been attributed to a bulk effect?
- Supplementary Fig. 3 – It is not possible to accurately deduce IC50 from the CP2 curve. Either re-do the experiment with a different concentration range or assume IC50 > than X. Also, the hill slope for CP2 seems implausible. CP5 and CP7 curves could do with more points at the higher concentration end.
- Supplementary Fig. 4 – How is data consistent with CP6 being uncompetitive i.e. this looks like it decreases binding of RGS5 substrate protein whereas the uncompetitive inhibitor would not change binding of the substrate (or increase it if not already saturated)?
- Supplementary Fig. 12 - define O as ornithine in legend
- Supplementary methods - Include chloroacetylation method in the general procedure A: Automated Peptide Synthesis (SYRO I peptide synthesiser)

Reviewer #3

(Remarks to the Author)

Reviewer #4

(Remarks to the Author)

Jiramongkol et al. "An mRNA-display derived cyclic peptide scaffold reveals the substrate binding interactions of an N-terminal cysteine oxidase"

The NCO family of oxygenase has an active site composed of an iron atom and its coordinating residues. Obtaining the complex structure with substrate has been challenging for several reasons: the substrate binding affinity to the enzyme is insufficient, leading to immediate enzymatic catalysis upon binding. Therefore, for structural studies require additional modifications, such as mutating active site residues without altering binding affinity or using tight-binding inhibitors to mimic the transition state of the enzymatic reaction mechanism.

The authors used the Random nonstandard Peptide Integrated Display (RaPID) method to screen for cyclic peptides that bind to human N-terminal cysteine oxidase (NCO; 2-aminothioethanol oxygenase, ADO) and determined their complex structures. They identified the cyclic peptide CP6, which exhibits nanomolar affinity to ADO, as a scaffold for elucidating the substrate analog-bound structure of ADO. The structural insights into the serine moiety of the substrate analog and its bidentate coordination with the active site cobalt, as well as the identification of Asp206 as a key catalytic residue, significantly enhance our understanding of ADO's function. Furthermore, their enzyme kinetic analyses of cyclic peptides and ADO mutants align well with the structural data, reinforcing their findings.

I believe that their use of high-affinity synthetic peptides to overcome ADO's low affinity for its native substrate could provide valuable tools for understanding the reaction mechanism of cysteine oxygenases across different species. This approach could also be applied to other proteins facing similar challenges.

I respectfully present a few questions and suggestions that may help clarify certain aspects of the manuscript:

<< Major comments >>

1. In a certain sense, the authors seem fortunate to have obtained cyclic peptides that effectively bind to and inhibit the target enzyme. I am particularly curious whether this RaPID method can be broadly applied to other enzymes, especially those within the NCO family. Although numerous examples demonstrate the use of this technique for discovering novel ligands (Kamalinia et al., Chem. Soc. Rev., 2021), the fact that all top 8 CPs identified in this study are ADO inhibitors is noteworthy. Upon reviewing the molecular surface structure of ADO, I observed multiple potential ligand-binding sites. It would be valuable if the authors could address in the Discussion section why such a high success rate was observed for ADO.

2. Concerning Supplementary Figure 8, additional evidence supporting the identification of the extra density near cobalt as "Tris" would be beneficial. While other apo structures (e.g., PDB IDs 8UAN and 8U9J; Li et al., JACS, 2024) also used Tris in the protein storage buffer (50 mM), they did not show similar extra density for Tris. Unfortunately, I was unable to compare your experimental conditions due to the lack of detailed information regarding the concentration of Tris buffer used for protein storage or crystallization in the Materials and Methods section. As described in the Figure, the authors proposed that Tris is present in the multiple conformations near the metal center of the ADO-CP6 complex. To strengthen this claim, it would be helpful to show each electron density map (1~4) with the different colors for negative and positive $F_o - F_c$ maps. If the bound-Tris is uncertain, the authors could consider softening the claim.

3. In Figure 2c and Supplementary Figure 9b, I noticed that CP6-F6S exhibits a lower IC_{50} compared to CP6, yet its enzymatic kinetics are similar to those of the DMSO control. This raises the question of whether CP6 act as a competitive inhibitor or through a different mode of action. Including enzymatic kinetics data for RGS5 at higher concentrations of CP6-F6S could help elucidate its mode of inhibition. Additionally, providing the direct kinetic parameters such as K_m and k_{cat} (or K_i) rather than K_D and IC would clarify the mode of inhibition.

4. The enzymatic reaction mechanisms of ADO and related thiol dioxygenases from previous studies are illustrated in Supplementary Figure 19. The authors align with the reaction mechanism proposed by recent structural studies of the cobalt(II) substituted ADO, supported by spectroscopic data, which suggests a non-high-valent catalytic pathway (Li et al., JACS, 2024). While the potential role of the Asp206 carboxylate during turnover is highlighted (Supplementary Figure 21), it remains unclear how the side chain of Asp206 functions within the reaction mechanism. For example, there is no indication of an interaction between the carboxylate of Asp206 and O_2 or H_2O in the proposed mechanism shown in Supplementary Figure 21. Clarifying this aspect would provide valuable insight into its catalytic role.

<< Minor comments >>

1. In the manuscript, you discuss the rotation of ADO-W257 (lines 220-222), but this rotation is not shown in Figure 2b or the Supplementary Figures. Including an illustration of this rotation (or at least an arrow) would enhance the reader's understanding and better support this point.

2. According to the ADO structure complexed with CP6-L8K-Ser and your description, the side chain orientation of ADO-D206 shifts to the conformation favorable for substrate oxidation by Nt-Ser from otherwise unfavorable conformation induced by CP6 binding. I wonder whether ADO could oxidize Nt-Cys, even with low efficiency, if CP6-L8K-Cys was used as a substrate.

3. The manuscript contains numerous sentence fragments throughout. While this style can sometimes be effective, it may disrupt the flow of the narrative (e.g., five instances of spacing between sentences on page 5).

4. In the Materials and Methods section, the authors detail ADO sample preparation for various experiments, such as RaPID/SPR experiments, enzyme assay, and X-ray crystallography. However, the description of ADO purification could be more concise or moved to the Supporting Information - Supplementary Methods section. Conversely, a more detailed description of the crystallization conditions would be beneficial for comparison with other groups' findings.

5. There is a typo on line 653: it should read "Supplementary Table 8" instead of "Supplementary Table 6".

6. In Supplementary Table 8, the term "other" under the number of atoms must be specified (e.g., metals and Tris atoms). Are there any other types of atoms included?

Reviewer #5

(Remarks to the Author)

Version 1:

Reviewer comments:

Reviewer #2

(Remarks to the Author)

We are happy that the authors have satisfactorily addressed our previous concerns.

Reviewer #3

(Remarks to the Author)

Reviewer #4

(Remarks to the Author)

I have checked that the authors have adequately revised my questions, and I am mostly satisfied with their responses.

Although some of their replies are not based on additional experiments, the final conclusion remains unchanged, and the text, including the Discussion, has been significantly improved.

Therefore, I believe the manuscript is now acceptable.

Reviewer #5

(Remarks to the Author)

Response to Reviewers Comments

Reviewer #1 (Remarks to the Author):

N-degron pathway dictates the half-life of a protein based on N-terminal amino acid residues, some of which are modified from translated side chains. Especially, N-terminal Cys is a tertiary destabilizing residue in the Arg/N-degron pathway and could be oxidized to secondary destabilizing residue by N-terminal cysteine oxidase (NCO), which enhances arginylation for further protein degradation processes. One of major problems in the Arg/N-degron pathway is that we do not know physiological protein substrates of enzymes that convert tertiary destabilizing residues to secondary destabilizing residues (NCO, NTAN1, and NTAQ1). Thus, this manuscript is of great importance in that the authors tried to find substrates of NCO using an mRNA-display derived cyclic peptide scaffold (RaPID)

Firstly, the authors identified and characterized cyclic peptide (CP) inhibitors of the NCO 2-aminoethanethiol dioxygenase (ADO) using RaPID. Eight most enriched CPs (CP1-8) were selected, produced by solid-phase peptide synthesis, and analyzed by in vitro interaction assay SPR using single cycle kinetic analysis. Interestingly, the authors could determine the crystal structure of ADO in complex with CP6 after careful comparison with other crystal structures of unbound ADO. With the CP6-bound ADO structure in hands, the authors could elucidate the substrate interacting modes and provide reasonable reaction mechanism models. In addition, CP6 was used as a scaffold to graft substrate moieties and some substrate analogues, including CP6-L8K-Ser and CP6-L8d-Gly-Ser, were selected for further kinetic analysis and structure determination. The first crystal structure of an ADO substrate complex could provide valuable structural information at an atomic level about reaction mechanism and rational inhibitor design.

Even though metal substituted ADO and substrate analogue residues were used, the authors tried their best to connect their findings to native/physiological interaction using currently available methods, such as comprehensive mutational studies and MD simulation. Overall, this manuscript is very well written and the experimental description in this manuscript was sufficiently and scientifically sound. This manuscript could be published in the journal with very minor revision as described in the below.

Thank you for your positive reception of our manuscript!

1. Introduction: The ADO reaction mechanism is described as texts very in detail. Even this reviewer could not follow completely without visual supports. Thus, please add a schematic diagram/figure about the reaction mechanism that the authors describe.

Thank you for highlighting this inconsistency. We have added reactions schemes for CDO (which were described in the introduction) to the Supporting Information of the revised manuscript. Please see Supplementary Figure 2. Reaction schemes for ADO can be found in Supplementary Figures 23-25.

2. Discussion: The fourth paragraph (Another difference ~~~~ this claim.) seems to be somewhat abrupt and to focus on a specific aspect of their findings with speculation and repetitive/iterative description with the Results section. This fourth paragraph

could be shortened and might be connected to the third paragraph in the Discussion section. Instead, the authors could discuss possible extension of their findings to the N-degron pathway, followed by the current discussion about its disease/agricultural relation (the fifth paragraph).

Thank you for suggesting this improvement. We have shortened and consolidated this section of the discussion in the revised manuscript. Please see lines 433-441.

3. Typo in Abstract: 2-amino(e)thanethiol dioxygenase (ADO).

Thank you for highlighting this error. It has been corrected in the revised manuscript. Please see line 34.

Reviewer #2 (Remarks to the Author):

Jiramongkol et. al describe the determination of the ADO crystal structure in complex with pseudo-substrate analogues. No crystal structure of ADO with substrates has been reported prior to this work. This is attributed to the rapid substrate turnover and low substrate affinity. The authors took advantage of the RaPID mRNA display approach to find cyclic peptide binders of ADO, which were then used as tools to graft substrate analogues of ADO bearing a pseudo-Nt-Cys or -Ser residue. This aided stabilisation of the substrate analogues and led to the determination of two crystal structures of ADO with different pseudo-substrates. This structural data provide evidence on the binding mode of ADO substrates, strongly pointing to a bidentate association of the substrate which can inform both reaction mechanism and inhibitor design. Further structure-guided mutagenesis experiments highlight the importance of D206 in the catalytic mechanism of ADO. Overall, this manuscript provides valuable structural information on the ADO substrate binding mechanism, including strong evidence for a bidentate binding mode – something that has been disputed to this point. More generally it also highlights the use of de novo cyclic peptides as substrate-grafting tools. On the whole the data is sound and well presented and the manuscript is very well written.

On the basis of this I anticipate it will be of great interest to a broad range of Nature Communication readers and recommend acceptance once the authors address the following minor issues:

Thank you for your positive reception of our manuscript!

- More experimental details on the RaPID selection process should be added to Supporting Information. For example, full DNA library sequences used for selection, comparison of enrichment for positive and negative selections, next-generation sequencing data.

Thank you for requesting additional information on the RaPID selection process.

- 1) We have added the oligonucleotide and primer sequences used for library generation and selection to the methods section of the revised manuscript. Please see line 573 and between lines 594 and 595.

- 2) We used qPCR to monitor the enrichment and recovery of positive and negative selections. However, it did not provide reliable information on selection progress. For your reference, we have provided the results below. We don't believe the data is a valuable addition to the manuscript and we would prefer not to include it.
- 3) We have added the top 100 sequences from selection round 3 and 6 (i.e. the final round), along with their corresponding frequencies, as Supplementary Data files in the revised manuscript.

- Define the competitive vs. uncompetitive inhibitor nomenclature used in the manuscript – by definition, an uncompetitive inhibitor binds to the enzymes-substrate complex, enhancing the K_m for the substrate whilst inhibiting activity but the competitive SPR e.g. in SF4 shows peptides still block substrate binding (in fact the “uncompetitive” peptide CP6 seems to be one of the most effective substrate competitors) and the crystal structure shows the peptide binding where the substrate has to bind suggesting it is unlikely the peptide can bind to the enzyme-substrate complex. Please expand on this.

Thank you for highlighting this discrepancy in inhibitor classification. We described CP6 and 8 as uncompetitive inhibitors based on the kinetic data, which demonstrates that they decrease both K_{cat} and K_m . However, we agree that the SPR and crystallographic data indicate that they (or, at the very least, CP6) are competitive in nature.

We have made this distinction in the revised manuscript, removing direct references of CP6 and 8 as uncompetitive inhibitors. Instead, we state that they exhibit uncompetitive properties, behaviours or characteristics according to their effect on enzyme kinetics.

This includes changes on lines 121, 184, 186, 229, and 250.

- Page 4 - provide a reference for the ADO activity assay used since it is an “established stopped assay”.

Thank you for highlighting this oversight. Three references (7, 35 and 36) have been added to the revised manuscript. These papers describe prior use of the assay. Please see line 167.

- Fig. 1D, Table 1, Supplementary Figures 3 and 9 and Supplementary Table 4 - provide errors in the IC50 values obtained.

Thank you for highlighting this oversight. We have added error to IC50 values in these figures and tables in the revised manuscript. Note – Supplementary Figures 3 and 9 are now Supplementary Figures 5 and 11.

- Page 5 and 7 – “root mean squared deviation (rmsd)” – RMSD should be in capital letters.

Thank you for highlighting this oversight. The abbreviation for root mean squared deviation has been capitalised in the revised manuscript. Please see lines 214 and 325.

- Page 6 – “In the presence of ADO-E92A, CP6 exhibited characteristics that more closely align with a competitive inhibitor, increasing the apparent K_m and k_{cat} relative to the DMSO control and wild-type ADO, respectively.” Based on Supplementary Figure 10, the k_{cat} decreases.

Thank you for highlighting this observation. While the k_{cat} of ADO-E92A does decrease in the presence of CP6, this decrease is less pronounced than wild-type ADO in the presence of CP6, which, in addition to the increase in apparent K_m , indicates that the mutation E92A makes CP6 exhibit kinetic properties that more closely align with a competitive inhibitor.

The text has been altered in the revised manuscript to clarify this distinction. Please see line 247.

- Page 6 – the authors’ produce two substrate analogues of CP6, CP6-L8K-Ser and CP6-L8K-Cys. They crystallised ADO in complex with CP6-L8K-Ser at 1.60 Å resolution, even though CP6-L8K-Cys appears to be a more physiologically-relevant substrate. Has the CP6-L8K-Cys analogue been challenging to crystallise? Please elaborate on this. Can the authors test whether the Cys variants are synthetic substrates for ADO?

Thank you for raising this interesting question. Unfortunately, you are correct. Despite multiple attempts, we were able to generate crystals of ADO in complex with CP6-L8K-Cys, even under the same crystallisation conditions as CP6-L8K-Ser.

Sadly, we are unable to test whether CP6-L8K-Cys is an ADO substrate, as we depleted our stock of peptide during crystallisation studies (and these drops have since dried out). However, we believe it could be a substrate, and this might be one of the reasons why we struggled to produce crystals, as ADO does not bind sulfinic

acid/sulfinate ¹. Alternatively, there may be issues with peptide dimerization as N-terminal cysteine (which has a low pKa) readily form disulfide bonds ².

We have highlighted these potential reasons for failing to generate crystals of ADO in complex with CP6-L8K-Cys in the discussion of the revised manuscript. Please see lines 391-393.

- Page 8 - typo “moities” – moieties

Thank you for highlighting this error. It has been corrected in the revised manuscript. Please see line 335.

- Page 12 – space missing between mRNA and libraries

Thank you for highlighting this error. It has been corrected in the revised manuscript. Please see lines 574-575.

- Page 12 - “To allow for genetic code reprogramming, a custom “Solution A” was used supplemented with 19 amino acids (-Met) and N-chloroacetyl-L-tyrosine aminoacylated initiator tRNA as previously described [REFS].” References missing.

Thank you for highlighting this oversight. An appropriate reference (Goto Y, Katoh T, Suga H. Flexizymes for genetic code reprogramming. *Nat Protoc* **6**, 779-790 (2011)) has now been added to the revised manuscript. Please see line 582.

- Page 12 – “After washing, the fused peptide–mRNA/cDNA was isolated from the beads by heating to 95 °C for 5 min, and cDNA was amplified by PCR, purified by ethanol precipitation and transcribed as above to produce the enriched mRNA library for the next round of selection.” What was the fused peptide-mRNA/cDNA washed with specifically?

Thank you for highlighting this oversight. Details for the wash buffer (Tris buffered saline (50 mM Tris pH 8.0, 150 mM NaCl) containing 0.05% tween-20 (TBS-T)) have been added to the methods section of the revised manuscript. Please see lines 587-588.

- Page 13 – which commercial crystallisation screens were used?

Thank you for highlighting this oversight. We have added the commercial crystallisation screens and final crystallisation conditions used to the methods section of the revised manuscript. Please see lines 633 and 640-643.

- Page 14 - “Ribosomal synthesis of the macrocyclic peptide library from the puromycin-linked mRNA library was performed using the PURExpress DRF kit (New England Biolabs) with RF2 and RF3 added.” Define D in DRF kit – delta?

Thank you for highlighting this error. D does correspond to delta. The appropriate symbol has been added in the revised manuscript. We have also added 123, in reference to the release factors omitted from the initial translation mix. This is the official name of the kit. Please see line 579.

- Provide pub ID of the crystal structures obtained.

Protein Data Bank Identification codes are provided in the main text, when each crystal structure is introduced (please see lines 201, 277, and 300), in the crystallography table (Supplementary Table 9), and the data availability section (please see lines 685-689). We are happy to provide these codes in additional sections of the manuscript if requested.

- The authors expressed and purified several ADO mutants, however SDS-PAGE gels are missing. They should be added to the Supporting Information along with the sequence of the parent ADO construct.

Thank you for highlighting this oversight. We have added the amino acid sequence and SDS-PAGE analysis of each construct and variant to the Supporting Information section of the revised manuscript. Please see Supplementary Figure 3.

- Fig. 2C – include concentration of peptide

Thank you for highlighting this oversight. Each CP was added at its IC₅₀ concentration to capture changes in substrate association and turnover during partial inactivation of the enzyme, following the same strategy used during initial analysis of the cyclic peptides. These details have been added to Figure Legend 2 in the revised manuscript.

- Fig. 2C – the data on CP6-F6S is confusing. CP6-F6S is more potent (by IC₅₀ and KD – SF9) than CP6 but the Michaelis-Menten kinetic plots show that it doesn't alter cat or Km? Please can the authors elaborate on this.

Thank you for asking for clarification on this data. As stated above, each CP was added at its IC₅₀ concentration to capture changes in substrate association and turnover during partial inactivation of the enzyme, following the same strategy used during initial analysis of the cyclic peptides. As a result, CP6, CP6-K4A and CP6-F6S were added at 1, 21 and 0.2 uM, respectively. At this concentration, the changes for CP6-F6S relative to the DMSO control are reasonably subtle according to the Michaelis-Menten plot, but large relative to CP6, with a significant increase in kcat.

We have repeated the experiment with higher concentrations of CP, adding CP at two times its IC₅₀ concentration. These results follow a similar trend for CP6-F6S (i.e. kcat increases relative to CP6), but the kinetic parameters for CP-K4A become undefined.

This data has been added to the Supporting Information section, and mentioned in the text, of the revised manuscript. Please see Supplementary Figure 12, Supplementary Table 5, and lines 243-244.

- Fig. 3C – I assume that d is 2,4-diaminobutyric acid? Please define in the figure legend

Thank you for highlighting this oversight. Dap refers to 2, 3-diaminopropionic acid. This detail has been added to Figure Legend 3 and main text in the revised manuscript. Please see line 294.

- Fig. 3 (legend) – typos; “ADO-206 also interacts with the water molecule trans to ADO-His112, which is the putative O₂ binding site.” Change to ADO-D206 and ADO-H112 (for consistency).

Thank you for highlighting this error. It has been corrected in the revised manuscript.

- Supplementary Fig. 2 – The derived K_D of CP2 and CP3 is 5-7x higher than the highest concentration of CP2 used in the experiment. Can the authors comment on how accurate they believe this K_D value? Also, can they comment on the accuracy of the CP8 fit which appears to have mostly been attributed to a bulk effect?

Thank you for highlighting these observations.

- 1) The K_D values of CP2 and 3 were calculated using off-rate kinetics, so we are confident that they are in the right ballpark, however, given the large error associated with these figures, we have listed the K_Ds for CP2 and 3 as >5 μM in the revised manuscript. Please see Supplementary Figure 4, Table 1 and line 158.
- 2) We have repeated the SPR experiment with CP8, using improved buffer matching to reduce the bulk effect. We have added this data to the revised manuscript and updated the necessary tables. As expected, the K_D is lower and less error prone. Please see Figure 4, Table 1 and line 157.

- Supplementary Fig. 3 – It is not possible to accurately deduce IC₅₀ from the CP2 curve. Either re-do the experiment with a different concentration range or assume IC₅₀ > than X. Also, the hill slope for CP2 seems implausible. CP5 and CP7 curves could do with more points at the higher concentration end.

Thank you for highlighting this oversight. We agree that we cannot deduce an accurate IC₅₀ value for CP2 using the data points collected. We have defined the IC₅₀ as >50 μM in the revised manuscript. We have also stated that the hill slopes for CP2 and 4 are undefined. Please see Supplementary Figure 5 and Table 1.

- Supplementary Fig. 4 – How is data consistent with CP6 being uncompetitive i.e. this looks like it decreases binding of RGS5 substrate protein whereas the uncompetitive inhibitor would not change binding of the substrate (or increase it if not already saturated)?

Thank you for highlighting this discrepancy in inhibitor classification. As stated above, we described CP6 as an uncompetitive inhibitor based on the kinetic data, which demonstrates that it decreases both K_{cat} and K_m. However, we agree that the SPR and crystallographic data indicate that it binds at the active site of the enzyme in the absence of substrate.

We have made this distinction in the revised manuscript, removing direct references of CP6 as an uncompetitive inhibitor. Instead, we state that it exhibits uncompetitive properties, behaviours or characteristics according to its effect on enzyme kinetics.

This includes changes on lines 121, 184, 186, 229, and 250.

- Supplementary Fig. 12 - define O as ornithine in legend

Thank you for highlighting this oversight. This definition has been added in the revised manuscript. Please see the legend of Supplementary Figure 15.

- Supplementary methods - Include chloroacetylation method in the general procedure A: Automated Peptide Synthesis (SYRO I peptide synthesiser)

Thank you for highlighting this oversight. Conditions for coupling chloroacetic acid have been added to General Procedure A in the Supplementary Method section in the revised manuscript.

Reviewer #3 (Remarks to the Author):

Reviewer #4 (Remarks to the Author):

Jiramongkol et al. “An mRNA-display derived cyclic peptide scaffold reveals the substrate binding interactions of an N-terminal cysteine oxidase”

The NCO family of oxygenase has an active site composed of an iron atom and its coordinating residues. Obtaining the complex structure with substrate has been challenging for several reasons: the substrate binding affinity to the enzyme is insufficient, leading to immediate enzymatic catalysis upon binding. Therefore, for structural studies require additional modifications, such as mutating active site residues without altering binding affinity or using tight-binding inhibitors to mimic the transition state of the enzymatic reaction mechanism.

The authors used the Random nonstandard Peptide Integrated Display (RaPID) method to screen for cyclic peptides that bind to human N-terminal cysteine oxidase (NCO; 2-aminothioethanol oxygenase, ADO) and determined their complex structures. They identified the cyclic peptide CP6, which exhibits nanomolar affinity to ADO, as a scaffold for elucidating the substrate analog-bound structure of ADO. The structural insights into the serine moiety of the substrate analog and its bidentate coordination with the active site cobalt, as well as the identification of Asp206 as a key catalytic residue, significantly enhance our understanding of ADO's function. Furthermore, their enzyme kinetic analyses of cyclic peptides and ADO mutants align well with the structural data, reinforcing their findings.

I believe that their use of high-affinity synthetic peptides to overcome ADO's low affinity for its native substrate could provide valuable tools for understanding the reaction mechanism of cysteine oxygenases across different species. This approach could also be applied to other proteins facing similar challenges.

Thank you for your positive reception of our manuscript!

I respectfully present a few questions and suggestions that may help clarify certain aspects of the manuscript:

<< Major comments >>

1. In a certain sense, the authors seem fortunate to have obtained cyclic peptides that effectively bind to and inhibit the target enzyme. I am particularly curious whether this RaPID method can be broadly applied to other enzymes, especially those within the NCO family. Although numerous examples demonstrate the use of this technique for discovering novel ligands (Kamalinia et al., Chem. Soc. Rev., 2021), the fact that all top 8 CPs identified in this study are ADO inhibitors is noteworthy. Upon reviewing the molecular surface structure of ADO, I observed multiple potential ligand-binding sites. It would be valuable if the authors could address in the Discussion section why such a high success rate was observed for ADO.

Thank you for raising this interesting observation. We agree that there is an aspect of luck in identifying peptides that bind and alter the function of a protein through RaPID/mRNA display. However, we would argue that –

- 1) appropriate library design and experimental set up can increase success, and,
- 2) ADO is an amenable target.

RaPID/mRNA display is often used to identify ligands of targets with undefined binding pockets, namely those that participate in protein-protein interactions, which is challenging. However, as you correctly point out, ADO contains multiple potential ligand-binding sites, which will facilitate interactions with the peptide library. The largest of these constitutes the active site, an open and functionally essential cavity, which will enhance the identification of high affinity binders with inhibitory properties.

That said,

- 1) CP2, 3 and 4 aren't good ligands or inhibitors (CP3 exhibits no inhibition properties) and their mode of action is undetermined (it is possible that they don't bind to the active site). And,
- 2) CP1, 5, 6, and 7 are related in sequence and represent a peptide family, rather than distinct inhibitors.

Together, we would argue that only 2 unique inhibitors were identified during this investigation: CP6 (and its related family members CP1, 5, and 7), and CP8.

Although interesting, we would prefer not to add these points to the discussion of the revised manuscript as we think it detracts from the take home messages of the study, which focuses on understanding how ADO works.

2. Concerning Supplementary Figure 8, additional evidence supporting the identification of the extra density near cobalt as “Tris” would be beneficial. While other apo structures (e.g., PDB IDs 8UAN and 8U9J; Li et al., JACS, 2024) also used Tris in the protein storage buffer (50 mM), they did not show similar extra density for Tris. Unfortunately, I was unable to compare your experimental conditions due to the lack of detailed information regarding the concentration of Tris buffer used for protein storage or crystallization in the Materials and Methods section. As described in the Figure, the authors proposed that Tris is present in the multiple conformations near the metal center of the ADO-CP6 complex. To strengthen this claim, it would be helpful to show each electron density map (1~4) with the different colors for negative and positive Fo-Fc maps. If the bound-Tris is uncertain, the authors could consider softening the claim.

First, we apologise for not including information on the crystallisation conditions. These details have been added to the methods section of the revised manuscript. Please see lines 631, 633, and 640-643. Tris was introduced during complex formation, as the cyclic peptide was dissolved in 50 mM Tris pH 7.5, 400 mM NaCl.

Second, thank you for highlighting this interesting difference. We speculate that CP6 influences metal coordination by:

- 1) Altering the conformation of ADO-D206, which eliminates a hydrogen bond with the water molecule occupying the putative O₂ binding site on the metal cofactor. This may lower the affinity of the water molecule, allowing it to be displaced by Tris. And,
- 2) Trapping Tris in the active site, increasing its local concentration and, therefore, encouraging an interaction with the metal centre.

We have added Fo-Fc maps for each arrangement of Tris to the Supporting Information section (please see Supplementary Figure 10), however, it is still difficult to conclude what the density corresponds to definitively. Tris was the most likely interpretation given the information available.

We have both strengthened and softened our claim in the revised manuscript by –

- 1) referencing a paper in which Tris has been observed interacting with the metal centre of an NCO (PCO2, PDB:7CXZ), and
- 2) stating that ‘other interpretations are possible’.

Please see lines 222-226.

3. In Figure 2c and Supplementary Figure 9b, I noticed that CP6-F6S exhibits a lower IC₅₀ compared to CP6, yet its enzymatic kinetics are similar to those of the DMSO control. This raises the question of whether CP6 act as a competitive inhibitor or through a different mode of action. Including enzymatic kinetics data for RGS5 at higher concentrations of CP6-F6S could help elucidate its mode of inhibition.

Additionally, providing the direct kinetic parameters such as K_m and k_{cat} (or K_i) rather than K_D and IC would clarify the mode of inhibition.

Thank you for asking for clarification on this data. Each CP was added at its IC_{50} concentration to capture changes in substrate association and turnover during partial inactivation of the enzyme, following the same strategy used during initial analysis of the cyclic peptides. These details have been added to Figure Legend 2 in the revised manuscript. As a result, CP6, CP6-K4A and CP6-F6S were added at 1, 21 and 0.2 μM , respectively. At this concentration, the changes for CP6-F6S relative to the DMSO control are reasonably subtle according to the Michaelis-Menten plot, but large relative to CP6, with a significant increase in k_{cat} .

We have repeated the experiment with higher concentrations of CP, adding CP at two times its IC_{50} concentration. These results follow a similar trend for CP6-F6S (i.e. k_{cat} increases relative to CP6), but the kinetic parameters for CP-K4A become undefined.

This data has been added to the Supporting Information section, and mentioned in the text, of the revised manuscript. Please see Supplementary Figure 12, Supplementary Table 5, and lines 243-244.

Kinetic parameters are presented in Supplementary Table 4. This has been referenced in the revised manuscript. Please see line 243.

4. The enzymatic reaction mechanisms of ADO and related thiol dioxygenases from previous studies are illustrated in Supplementary Figure 19. The authors align with the reaction mechanism proposed by recent structural studies of the cobalt(II) substituted ADO, supported by spectroscopic data, which suggests a non-high-valent catalytic pathway (Li et al., JACS, 2024). While the potential role of the Asp206 carboxylate during turnover is highlighted (Supplementary Figure 21), it remains unclear how the side chain of Asp206 functions within the reaction mechanism. For example, there is no indication of an interaction between the carboxylate of Asp206 and O_2 or H_2O in the proposed mechanism shown in Supplementary Figure 21. Clarifying this aspect would provide valuable insight into its catalytic role.

Thank you for requesting clarification on the role of Asp206 during ADO turnover. We highlighted a number of potential roles for Asp206 in the discussion of the initial manuscript, one of which was superoxo orientation. Here, Asp206 indirectly interacts with the superoxo species through charge repulsion, positioning the distal oxygen away from the thiol group. In this scenario, Asp206 does not directly contribute to the mechanism. It simply aligns the superoxo species. We believe this is still a viable contribution.

For clarity and consistency, we have updated the text (please see lines 422-432) and added reaction schemes for each role discussed in the revised manuscript. This includes orientating and directing the superoxo group (Supplementary Figure 24A and 24B), stabilising a reactive oxygen intermediate (Supplementary Figure 25A), and facilitating O_2 exchange (Supplementary Figure 25B).

<< Minor comments >>

1. In the manuscript, you discuss the rotation of ADO-W257 (lines 220-222), but this rotation is not shown in Figure 2b or the Supplementary Figures. Including an illustration of this rotation (or at least an arrow) would enhance the reader's understanding and better support this point.

Thank you for highlighting this oversight. We have added a figure showing the conformation of ADO-W257 in the presence and absence of CP6 in the revised manuscript. Please see Supplementary Figure 9.

2. According to the ADO structure complexed with CP6-L8K-Ser and your description, the side chain orientation of ADO-D206 shifts to the conformation favorable for substrate oxidation by Nt-Ser from otherwise unfavorable conformation induced by CP6 binding. I wonder whether ADO could oxidize Nt-Cys, even with low efficiency, if CP6-L8K-Cys was used as a substrate.

Thank you for raising this interesting question. Sadly, we are unable to test whether CP6-L8K-Cys is a substrate, as we depleted our stock of peptide during crystallisation studies (and these drops have since dried out). However, we believe it could be a substrate, and this might be one of the reasons why we struggled to produce crystals, as ADO does not bind sulfinic acid/sulfinate ¹ (alternatively, there may be issues with peptide dimerization as N-terminal cysteine (which has a low pKa) readily form disulfide bonds ²).

We have highlighted this point in the discussion of the revised manuscript. Please see lines 391-393.

3. The manuscript contains numerous sentence fragments throughout. While this style can sometimes be effective, it may disrupt the flow of the narrative (e.g., five instances of spacing between sentences on page 5).

Thank you for suggesting this improvement. Where possible, we have consolidated paragraphs in the revised manuscript to improve the narrative.

4. In the Materials and Methods section, the authors detail ADO sample preparation for various experiments, such as RaPID/SPR experiments, enzyme assay, and X-ray crystallography. However, the description of ADO purification could be more concise or moved to the Supporting Information - Supplementary Methods section. Conversely, a more detailed description of the crystallization conditions would be beneficial for comparison with other groups' findings.

Thank you for highlighting this oversight. We have added the commercial crystallisation screens and final crystallisation conditions used to the methods section of the revised manuscript. Please see lines 631, 633, and 640-643.

5. There is a typo on line 653: it should read "Supplementary Table 8" instead of "Supplementary Table 6".

Thank you for highlighting this error. It has been corrected in the revised manuscript. Please see line 656.

6. In Supplementary Table 8, the term “other” under the number of atoms must be specified (e.g., metals and Tris atoms). Are there any other types of atoms included?

Thank you for highlighting this oversight. We have listed what ‘other’ corresponds to in the revised manuscript. Please see the note inserted under Supplementary Table 9 in the Supporting information.

Reviewer #5 (Remarks to the Author):

References

- 1) Patel K, *et al.* The enzymatic oxygen sensor cysteamine dioxygenase binds its protein substrates through their N-termini. *J Biol Chem* **300**, 107653 (2024).
- 2) White MD, Kamps J, East S, Taylor Kearney LJ, Flashman E. The plant cysteine oxidases from *Arabidopsis thaliana* are kinetically tailored to act as oxygen sensors. *J Biol Chem* **293**, 11786-11795 (2018).